

# Effect of ecological restoration programs on dust pollution in North China Plain: a case study

Xin Long[1,2], Xuexi Tie[1,2,3,4,5*], Guohui Li[1,2*], Junji Cao[1,2], Tian Feng[1,6], Li Xing[1,2], Zhisheng An[2]

[1]Key Laboratory of Aerosol Chemistry and Physics, Institute of Earth Environment, Chinese Academy of Sciences, Xi'an 710061, China

[2]State Key Laboratory of Loess and Quaternary Geology, Institute of Earth Environment, Chinese Academy of Sciences, Xi'an 710061, China

[3]Center for Excellence in Urban Atmospheric Environment, Institute of Urban Environment, Chinese Academy of Sciences, Xiamen 361021, China

[4]Shanghai Key Laboratory of Meteorology and Health, Shanghai, 200030, China

[5]National Center for Atmospheric Research, Boulder, CO 80303, USA

[6]Xi'an AMS Center, Xi'an 710061, China

*Correspondence and requests for materials should be addressed to:*

Xuexi Tie (email: xxtie@ucar.edu) *and* Guohui Li (email: ligh@ieecas.cn)





**Abstract:** In recent years, Chinese government has taken great efforts in initiating large-scale ecological restoration programs (ERPs) to reduce the dust pollutions in China. Using a satellite measurement product of Moderate Resolution Imaging Spectroradiometer (MODIS), the changes in land cover are quantitatively evaluated in this study. We find that grass and forest are increased in berried lands and deserts in northwestern China, which locate in the upwind regions of the populated areas of the North China Plain (NCP) in eastern China. As a result, the changes in land cover could produce important impacts on the dust pollutions in eastern of China. To assess the effect of ERPs on dust pollutions, a regional transport/dust model (WRF-DUST, Weather Research and Forecast model with dust) is applied to investigate the evolution of dust pollutions during a strong dust episode (from 2 to 8 March 2016). The calculations are intensively evaluated by comparing with the measured data. Despite some model biases, the WRF-DUST model reasonably reproduced the temporal variations and spatial distributions during the dust storm event. The correlation coefficient (R) between the calculated and measured dust concentrations is 0.77. The indices of agreement (IOAs) are 0.96 and 0.83, and the normalized mean bias (NMBs) are 2% and -15% in the dust source region (DSR) and the downwind populated area of NCP, respectively, suggesting that the WRF-DUST model well captures the spatial variations and temporal evolutions of the dust storm event. The impacts of EPRs induced land cover changes on the dust pollutions in NCP are quantitatively assessed using the WRF-DUST model. We find that the ERPs significantly reduce the dust pollutions in NCP, especially in the heart area of NCP (BTH, Beijing-Tianjin-Hebei). During the episode when the dust storm was transported from the DSR to NCP, the reduction of dust pollutions induced by ERPs ranges from -5% to -15% in NCP, with the maximum reduction of -15.3% (-21.0 μg m$^{-3}$) in BTH, and -6.2% (-9.3 μg m$^{-3}$) in NCP. Because the air pollution is severe in eastern China, especially in NCP, the reduction of dust pollutions has important effects on the severe air pollutions. This study shows that ERPs help to reduce air pollutions in the region, especially in springtime, suggesting the important contributions of ERPs to the air pollution control in China.

**Key words:** Ecological restoration programs; Dust pollution; North China Plain; WRF-DUST




## 1 Introduction


Dust particles have wide impacts on the Earth's radiative forcing budget (Liao et al.,
2004; Haywood et al., 2005), cloud formation (Rosenfeld et al., 2001), atmospheric
dynamics (Evan et al., 2008), air quality (Giannadaki et al., 2014), and ocean
biogeochemistry (Jickells et al., 2005) in various spatial and temporal scales.
Distinguished from the increasing trends observed in other major dust source regions
(Moulin et al., 1997), the East Asian dust storms are in decreasing trends since 1950s
except for a spike in dust activity (Lee and Sohn, 2011; Wang et al., 2017). The East
Asian dust storms could be transported to southern/eastern China (Qian et al., 2002),
Korea (Park and In, 2003), and Japan (Watanabe et al., 2014) and even the west coast
of North America (Cottle et al., 2013; Yoon et al., 2017). There are two dominant
source regions of East Asian dust storms locate in China, including the Taklamakan
Desert in northwest China and the Gobi Desert in Mongolia and northern China (Sun
et al., 2005; Wang et al., 2011). Along the transport pathway, mineral dust particles
lead to significant impacts on human's life in the densely populated areas of
southern/eastern part of China (Bian et al., 2011; Zhao et al., 2013).
To reduce dust pollution problem and to improve the environmental conditions, the
Chinese government has taken great efforts in initiating large-scale ecological
restoration programs (ERPs) (Yin and Yin, 2010; Cao et al., 2011). Chinese ERPs are
among the biggest programs in the world because of their ambitious goals, massive
scales, huge payments and potentially enormous impacts. As a result, the "Green Wall
of China" has been established in North China (Duan et al., 2011). There are strong
evidences that a remarkable vegetation increase trend has occurred in the dominant
dust source areas, northwestern China, especially after 2000 (Piao et al., 2003; Peng et
al., 2011). And the dust storm frequency in Northern China is generally in decreasing
trends (Li et al., 2014; Wang et al., 2007). However, it is still prevalent the ongoing
debate about effectiveness of the national ERPs. Numerous experts and government
officials have attributed the decrease trend to the success of ERPs in controlling dust
storms and combated desertification (Wang et al., 2007; Liu et al., 2008; Tan and Li,
2015). Conversely, several experts have doubted the program's effectiveness (Jiang,
2005; Wang et al., 2010; Cao et al., 2011), generally asserting the climate factors
being the main cause for the observed decrease of dust storms in northern China (Li et



al., 2014; Fan et al., 2017). Some experts further highlight the potential deterioration
of the ecosystem with severe depletion of soil moisture, especially in semiarid and
arid regions (Deng et al., 2016; Lu et al., 2016). Hence, there is an increasing need to
evaluate the China's ERPs at controlling dust pollution, particularly for the downwind
densely populated areas, to improve the decision support for ecological planning and
implementation. The mineral dust particles can also serve as carriers and reaction
platforms, and the heterogeneous dust chemistry may change the photochemistry, acid
deposition, and production of secondary aerosols in the atmosphere (Lou et al., 2014;
Fu, 2016; Zhou et al., 2016). The rigorous evaluation of ecological efforts is also
beneficial to improve the understanding of the attractive haze pollution research in
NCP.
There are difficulties to estimate the effectiveness of ERPs in dust control, which are
seldom quantitatively specified. On one hand, it is hard to quantify the influence of
ERPs in regional scale. The vegetation indices (e.g. NDVI, normalized-difference
vegetation index) are the most utilized parameters to conduct quantitative evaluation
of ERPs' effectiveness (Duan et al., 2011; Lü et al., 2015; Tan and Li, 2015). But the
vegetation indices are not efficient indicators for dust emission, which are mainly
related to erodibility of barren land surface directly (Bian et al., 2011). On the other
hand, it is hard to distinguish the influences of climate factors, which have been
generally asserted to be one of the main causes for the observed decrease of dust
storms in northern China. To exclude the influences of climate factors, Tan and Li
(2015) have compared the correlation of dust storm indices (intensity and frequency),
NDVI, wind speed, and precipitation within and outside the "Green Great Wall"
regions, qualitatively inferring the effectiveness of ERPs in reducing dust storm
intensity. However, the previous studies didn't quantify the roles of ERPs, such as the
detailed variations of ERPs, the effect of regional transport to downwind regions, etc.
The focus of our work is to use detailed satellite measurements to assess the region of
ERPs, and to use a regional model to quantify the effect of ERPs on the downwind
regions, especially in the NCP region.
Here our narrative is independently based on first-hand sources of satellite
measurements and WRF-DUST model simulation. We investigated the ERPs induced
land cover changes in China using the long-term MODIS land cover products. The



impacts of the ERPs induced land cover changes on the dust pollution in NCP were
further quantitatively evaluated using the WRF-DUST model. We selected two
regions of interest (ROIs) (**Fig. 1**): (1) the polluted and dense populated downwind
areas of dust storms, the North China Plain (NCP), including five provinces of the
Beijing, Tianjin, Hebei, Henan and Shandong; (2) the dust source region and
surrounding areas (DSR), including five provinces in the northwest of NCP (Ningxia,
Gansu, Shanxi, Inner Mongolia and Shanxi). The details of ROIs are shown in **Fig. 1b**.
The methodology and WRF-DUST model configuration are described in Sect. 2. Data
analysis and model results are presented in Sect. 3, together with the conclusions and
discussions in Sect. 4.
**2 Model and methodology**
**2.1  Dust pollutants measurements**
The China Ministry of Environmental Protection (China MEP) has commenced to
release real-time hourly observations of pollutants since 2013, including $O_3$, $NO_2$, CO,
$SO_2$, $PM_{2.5}$, and $PM_{10}$ (particulate matter with aerodynamic diameter less than 2.5 and
10 μm, respectively). We collected the hourly near-surface $PM_{2.5}$ and $PM_{10}$ mass
concentrations from the China MEP (http://www.aqistudy.cn/). Because there are no
detailed aerosol compositions measurements, the $PM_{2.5-10}$ (particulate matter with
aerodynamic diameter between 2.5 and 10 μm) mass concentrations (defined as
"[PMC]" in the later discussion) were utilized to analyze the dust pollution events.
According to several previous studies, the use of [PMC] also has two advantages: (1)
the size distribution of dust mass is center on the coarse model, and (2) the difference
between $PM_{10}$ and $PM_{2.5}$ can effectively decrease the uncertainty of anthropogenic
fine particulate matter, such as sulfate, nitrate, and organic aerosols (Ho et al., 2003;
Shen et al., 2011). A total of 184 cities (489 measurement sites) had [PMC]
observations in the research domain, including 30 cities within the DSR region and 53
cities within the NCP region (**Fig. 1a**). Because the prevailing winds were dominated
by west winds, the most measurement sites (as shown in **Fig. 1a**) locate in the
downwind area of the dust source regions (such as barren lands and deserts). As a
result, the China MEP measurement network provides a good opportunity to explore



the dust pollution evolution.

**2.2 MCD12Q1 data assimilation and land cover changes assessment**

We quantitatively evaluated the characteristics of annual land cover using the MODIS
land cover products (MCD12Q1), derived from the Terra- and Aqua- Moderate
Resolution Imaging Spectroradiometer (MODIS) observations (Friedl et al., 2002).
The MCD12Q1 have been widely used in studies of atmospheric science, hydrology,
ecology, and land change science (Gerten et al., 2004; Guenther, 2006; Reichstein et
al., 2007; Turner et al., 2007). The IGBP (International Geosphere Biosphere
Programme) classification within MCD12Q1 (Version 5.1) was analyzed to explore
the variability of the land use fraction (LUF) from 2001 to 2013. The IGBP layer is
generated using a supervised classification algorithm in conjunction with a revised
database of high quality land cover training sites (Friedl et al., 2010). Its accuracy is
estimated to be 72.3-77.4% (average 75%) globally, with a 95% confidence interval
(Friedl et al., 2002; Friedl et al., 2010).
The IGBP layer in MCD12Q1 is well consistent with the MODIS land use scheme in
the WRF-CHEM model, including 11 natural vegetation classes, 3 developed and
mosaicked land classes, and 3 non-vegetated land classes. **Supplementary Table 1**
shows the land use categories for the WRF-CHEM MODIS data and MCD12Q1. We
conducted the geospatial processing to assimilate the MCD12Q1 data (500 m) to fit in
the WRF-CHEM model (9 km in the present study) by the following steps. (1)
Convert the original raster MCD12Q1 dataset to vector files (esri-shapefile) and
re-projected them based on the geographic coordinate system configurations in
WRF-CHEM module (Its pre-processors). (2) Create vector files (shapefile) of each
grid based the domain of WRF-CHEM. (3) Access and iterate the selected
grid-shapefile to partition the converted the MCD12Q1 vector dataset into model
resolution using the Esri ArcGIS library (arcpy). The empty grids were populated
with the vector MCD12Q1 dataset using a spatial join operation in ArcGIS, joining
one input feature to one output feature (no aggregation) whenever input and output
polygon features intersect. This methodology preserves the values of the original
MCD12Q1 dataset. (4) Transcribe the newly merged and re-gridded MCD12Q1
datasets into text files readable in WRF-CHEM pre-processors, calculating the



gridded LUF of each category by
$LUF_{i,j,k} = \frac{Area_{i,j,k}}{Area_{i,j}}$                                    (1)
where $i$ and $j$ are grid indices, $Area_{i,j,k}$ stands for the total area of each land use
category $k$ within grid cell ($i, j$), and $Area_{i,j}$ is the area of grid cell ($i, j$). The $LUF_{i,j,k}$
ranges from 0 to 1, representing the emission potential of the specified dust source ($k$)
in each grid cell ($i, j$). The larger $LUF_{i,j,k}$, the higher dust emission potential.
**2.3  WRF-DUST model and configurations**
In the present study, we utilized a specific WRF-DUST model developed based on a
regional chemical model WRF-CHEM (version 3.2) (Grell et al., 2005). The
GOCART (Georgia Tech/Goddard Global Ozone Chemistry Aerosol Radiation and
Transport model) scheme (Chin et al., 2000) was utilized to calculate the physical
processes of dust emission, transport, dry depositions, and gravitational settling. The
dust particle sizes are divided into five size bins with effective radius of 0.7, 1.4, 2.4,
4.5 and 8.0 μm. The dust emission in each dust size bins is size-resolved. Dust
emission is dependent on the surface wind velocity (Ginoux et al., 2001), and surface
land cover properties (such as soil composition, vegetation, soil moisture content, and
soil erodibility) (Grini et al., 2005; Li et al., 2016a), which can be calculated by
$G_p = \begin{cases} C\gamma_p EV^2(V - Vt_p) & V > Vt_p \\ 0 & V \le Vt_p \end{cases}$                (2)
Where $G$ is the dust emission flux (kg s$^{-1}$); $p$ is the dust size bin; $C$ is a dimensional
factor (0.8 μg s$^2$ m$^{-5}$); $\gamma$ is the dust particle fraction; $E$ is the probability soil erosion
factor; $V$ is the near-surface wind velocity at 10 m (m s$^{-1}$); and $Vt$ is the threshold
velocity (m s$^{-1}$).
The WRF- DUST model was applied to simulate dust storm events in several
previous studies (Kang et al., 2009; Bian et al., 2011; Wang et al., 2012; Li et al.,
2016a). These studies reported that the WRF-DUST model is generally capable of





simulating dust storm events in the Asian region.
Because the dust emissions are strongly dependent on different categories of land
cover, to better investigate the impacts of land cover changes on the dust emission, we
modified the GOCART dust emission scheme, considering the each land cover dust
source categories other than the dominant category. The flux of dust emission G in
each grid is given by
$$G_p = \begin{cases} \sum_k LUF_k C\gamma_p E V^2 (V - V_p) & V > Vt_p \\ 0 & V \le Vt_p \end{cases} \tag{3}$$

$LUF_k$ denotes the gridded area fraction of land cover category $k$ derived from the
satellite data (MCD12Q1) assimilation. The other parameters are the same as those in
Eq. (2). We set the erosion factor E=0.12 for cropland and E=0.5 for barren following
the previous studies.
A dust storms episode from 2 to 8 March 2016 in northern China was simulated using
the WRF-DUST model. The WRF-DUST model adopts one grid with horizontal
resolution of 9 km centered in (112°E, 41°N) and 35 sigma levels in the vertical
direction. The grid cells used for the domain are 500×300 (**Fig. 1**). The physical
parameterizations include the microphysics scheme of Hong and Lim (2016), the
Mellor–Yamada–Janjic (MYJ) turbulent kinetic energy (TKE) planetary boundary
layer scheme (Janić, 2001), the unified Noah land-surface model (Chen and Dudhia,
2001). Meteorological initial and boundary conditions were taken from the 1°×1°
reanalysis data of National Centers for Environmental Prediction (NCEP). For the
episode simulations, the spin-up time is 3 days. Considering the impacts of the local
dust emission, the coarse mode of anthropogenic particulate matter emission was
included in the calculation. The detailed emission inventory was obtained from the
Multi-resolution Emission Inventory for China (MEIC) (Zhang et al., 2009), which is
then updated and improved for the year 2010 (http://www.meicmodel.org).
**2.4  Statistical methods for comparisons**
In order to assess the effect of the ERPs induced land cover changes on the dust
pollutions in China, the model calculation is statistically evaluated. The following
statistical parameters are calculated for evaluating the model calculation, including



the normalized mean bias (*MB*), the index of agreement (*IOA*), and the correlation
coefficient (R). These parameters are utilized to assess the WRF-CHEM model
performance in simulating air pollutants against measurements.
$$NMB = \frac{\sum_{i=1}^{N}(P_i - O_i)}{\sum_{i=1}^{N} O_i}$$   (4)
$$IOA = 1 - \frac{\sum_{i=1}^{N}(P_i - O_i)^2}{\sum_{i=1}^{N}(|P_i - \bar{O}| + |O_i - \bar{O}|)^2}$$   (5)
$$R = \frac{\sum_{i=1}^{N}(P_i - \bar{P})(O_i - \bar{O})}{[\sum_{i=1}^{N}(P_i - \bar{P})^2 \sum_{i=1}^{N}(O_i - \bar{O})^2]^{\frac{1}{2}}}$$   (6)
where $P_i$ and $O_i$ are the calculated and observed PMC concentrations ([PMC]),
respectively. $N$ is the total number of the predictions used for comparisons, and $\bar{O}$
represents the average of the prediction and observation, respectively. The *IOA* ranges
from 0 to 1, with 1 showing perfect agreement of the prediction with the observation.
The *R* ranges from -1 to 1, with 1 implicating perfect spatial consistency of
observation and prediction.
**3 Results and discussions**
**3.1 Land Cover change induced by ERPs**
The land surface changes due to the ecological restoration programs (ERPs) were
assessed using the MCD12Q1 product. From 2001 to 2013, the land cover exhibits
two obvious vegetation increase trends between the dust source region in northwester
China and dense populated areas in eastern China. Firstly, there is a regional
grass/savanna increase trend with obvious LUF increase of grass/savanna categories
(**Fig. 2b**), corresponding with a regional LUF decrease in barren categories in
northwestern China (**Fig. 2a**). The result is consistent with the previous research
based on long-term official and synthesized data, which also found a decreasing trend
of soil erosion areas in four provinces (e.g. Inner Mongolia, Gansu, Qinghai, and
Xinjiang), especially after 2000 (Zhang et al., 2016). Secondly, a regional forest LUF
increase trend occurs in the northwestern NCP (**Fig. 2c**), which agrees with the
previous study of Li et al., (2016b), who reported a remarkable forest growth in the



northwest of NCP from 2000 to 2010. As a result, two obvious vegetation protective
barriers arise throughout in southwest to northeast direction, which is well known as
"the Green Great Wall" with expectations to prevent the eastern of China from dust
pollution (Parungo et al., 1994; Liu et al., 2008; Cao et al., 2011).
The land cover changes, especially the obvious vegetation growths, are mainly caused
by the China's national ERPs. (1) The grassland increase is mainly induced by the
desertification control programs of the "Desertification Combating around Beijing
and Tianjin (DCBT)" and " the Shelterbelt Network Development Program (SNDP)".
They share with the goal of dust control, planning to protect grasslands and to convert
the desertified land into forestland and grassland. (2) The forest increase can be
attributed to many national afforestation programs, such as the "Natural Forest
Protection Program (NEPP)", "Grain for Green Project", "Three Norths Shelter Forest
System Project" and so on. The China's State Forestry Administration presented
enthusiasm to plant trees in the ecological restoration (Yin and Yin, 2010; Cao et al.,

269   2011).

**3.2  Model performance**
The hourly measurements of [PMC] in both the dust source region (DSR) and the
downward populated region (NCP region) were used to validate the WRF-DUST
model simulations. **Figure 3** presents the diurnal variations of calculated and
observed near-surface [PMC] averaged over the ambient monitoring site in provinces
within DSR and NCP. The model reasonably well reproduces the temporal variations
of surface [PMC] compared to the observations. e.g. the dust storm outbreak with
peak [PMC] in DSR are reasonably earlier than that in the downwind NCP areas. The
peak [PMC] occurred on 4 March within DSR (**Fig. 3a**), whereas occurred on 5
March within NCP (**Fig. 3b**). In the DSR region, the calculated results show a same
phase of the peak value compared with the measured peak on 4 March. However, the
calculated peak values show some underestimates of the measured value. In the NCP
region, the calculated results show a same phase of the peak value compared with the
measured peak on 5 March. The calculated peak value is similar to the measured peak.
However, after the peak value (after 6[th] March), the calculated results underestimate
of the measured value.





In the different provinces of the dust source region, the hourly provincial average
[PMC] can exceed 500 μg m$^{-3}$ in Ningxia, Gansu, and Inner Mongolia (the locations
of these provincial average show in **Fig. 1b**) before 20:00 4$^{th}$ March, implicating dust
storm outbreak in DSR. In the different provinces of the downward region, the peak
values have time-lags (hours to half day) compared to the peak values in DSR. For
example, the peak [PMC] arose first in Beijing with a time-lag of 7 hours. In other
four provinces of NCP (the locations of these provincial average show in Fig. 1b), the
time-lags are about 12 hours (**Fig. 3b**).
The statistical results show that the model generally exhibits good performance in
simulating [PMC] in the DSR region, involving IOA of 0.96 and NMB of 2% for
DSR. For the related provinces, all the IOAs exceed 0.85 and absolute NMB are
lower than 13% (**Fig. 3a1–5**). The model also generally reproduces the observed
[PMC] in NCP, with IOA of 0.83 for NCP and IOAs exceeding 0.67 for related
provinces. However, the model biases still exist, considerably underestimation biases
occurred on 6–7 March in NCP. The model underestimates considerably the observed
[PMC] with average NMB of -15% in NCP (**Fig. 3b0**). And the model cannot well
predict the observed [PMC] in Tianjin (**Fig. 3b2**), which is affected by the sea breeze
when the large-scale wind fields are weak (**Fig. 5e, 5h**). In general, however, current
numerical weather prediction models, even in research mode, still have difficulties in
producing the location, timing, depth, and intensity of the sea-breeze front (Banta et
al., 2005; Wang et al., 2013). The model reasonable predicts the [PMC] variations in
other four provinces in NCP, with IOAs more than 0.77, but with underestimation of
MBs varying from -25% to -3% (**Fig. 3b1, b3–5**), showing model biases in modeling
precipitation processes.
The episode-averaged calculation was compared with the measured result in **Fig. 4.**
Figure 4a provides the horizontal distributions of the simulated and the observed
near-surface [PMC], along with the simulated wind fields. The WRF-DUST model
reasonably reproduces spatial variation of [PMC] during the dust episode. The model
simulation is also able to provide a more detailed horizontal distribution, while the
measured data is generally lack of the data in the remote desert area (see **Fig. 4a**). The
correlation coefficient (R) between the simulations and observations is 0.77 (see **Fig.**
**4b**), suggesting that the model simulation is able to represent the measured result



during the dust episode period.
In order to evaluate the detailed temporal evolution of the dust plume, the daily
average calculated and measured dust distributions are shown in **Fig. 5**. On 2 March,
it was a starting dust storm stage, and both the observed and simulated [PMC] reached
as high as 200–300 μg m$^{-3}$ in the upwind DSR region, while in the downwind NCP
region, the concentrations of [PMC] were low, being only 20–50 μg m$^{-3}$ (**Fig 5a**). On
3 March, the dust storm was strengthened in the upwind DSR region (**Fig 5b**). On 4
March, the dust storm was further strengthened in the upwind DSR region. The area
of the dust storm in DSR was enlarged, and the concentrations of [PMC] were the
highest values of the episode, reaching to 300–500 μg m$^{-3}$. In addition, there were
strong northwest winds (> 10 m s$^{-1}$). Due to the strong northwest prevailing winds, the
dust storm started to be transported from upwind DSR to downwind NCP with
northwest to southeast direction (**Fig 5c**). On 5 March, due to the strong northwest
prevailing winds in the previous day, the dust storm reached to the NCP region, and
caused a remarkable [PMC] increase, with the concentrations rise to 100–200 μg m$^{-3}$.
At the same time, the dust plume was dispersed in DSR, showing a significant
decrease in [PMC]. The model results well represented these important feathers (**Fig**
**5d**). On 6–7 March, the dust storm passed through and the wind speed slowed down,
the [PMC] significantly decreased in both the DSR and NCP regions (**Fig. 5e-f**). The
correlation coefficients between measured and simulated [PMC] are 0.58–0.90 in
starting stage of the dust storm (**Fig. 5a-5c**), and 0.62 – 0.73 in the later stage of the
dust storm (**Fig. 5d-5f**).
Generally, the WRF-DUST model well captures the spatial variations and temporal
evolutions of dust storm during the episode. However, some model biases exist. For
example, the model underestimates the observed [PMC] in NCP, especially during the
later stage of the episode on 6–7 March (**Fig. 5e-f**), suggesting that several bias in the
model (such as the bias in meteorological simulation, a faster deposition, etc.) (Bian et
al., 2011; Duan et al., 2011; Bei et al., 2012).
**3.3  Effect of ecological restoration on dust pollution**
The evaluation the model simulation during dust storm episode suggests that the
WRF-DUST model is able to simulate the dust transport from the source region to



downwind areas, which can be used to assess the effect of ecological restoration on
the dust pollution in the populated region, such as NCP.
Despite the ongoing debate about effectiveness (Jiang, 2005; Liu et al., 2008; Wang et
al., 2010; Tan and Li, 2015; Deng et al., 2016), there are incontestably great changes
of the surface properties induced by the China's national ERPs (Yin and Yin, 2010;
Cao et al., 2011; Duan et al., 2011). We conducted model sensitivity studies to
quantitatively evaluate the impacts of the land cover changes on the dust pollution in
NCP. Two model simulations were performed. In the base case, the MCD12Q1
product with IGBP scheme in 2013 was utilized to represent the land cover situations
after the ERPs. Compared with the base simulation, another simulation was conducted
with same configuration and input data, except the land cover situations assimilated
from the MCD12Q1 product with IGBP scheme in 2001. This model simulation
represents the land cover situations without the effects of ERPs. The differences of the
two model simulations of dust concentrations were compared.
**Figure 6** presents the near-surface [PMC] change from 2001 to 2013, including the
temporal variations and the episode-average spatial variations. The vegetation
increase regions and downwind areas denoted the most remarkable change with [PMC]
reduction exceeding 20%, especially for the areas where barren converted to grassland
(**Fig. 2a, 2b, Fig. 6b, 6d**). The ERPs generally reduce the dust pollution in NCP
during the dust storm episode, except in Henan province. The episode-average [PMC]
reduction is -10% to -2% in the heart of NCP (BTH; Beijing, Tianjin, and Hebei) and
Shandong. In northern Hebei, the episode-reduce [PMC] can reach as high as -20% to
-10% (**Fig. 6b, 6d**). The changes of [PMC] are generally negative, implicating the
effectiveness of ERPs in preventing the dust pollution in NCP, especially for BTH.
During the episode when the dust storm was transported from the DSR to NCP, the
benefits of ERPs induced dust pollution reduction are remarkable, with the reduction
of [PMC] ranging -5% to -15% in NCP. The highest reduction of [PMC] induced by
ERPs are -15.3% (-21.0 μg m$^{-3}$) for BTH and -6.2% (-9.34 μg m$^{-3}$) for NCP (**Fig 6a,**
**6c**).
**Figure 7** shows the detailed horizontal distributions in the different stages of the
episode, such as T1 (08:00, 4 March), T2 (02:00, 5 March), T3 (13:00 5 March), and



T4 (04:00, 6 March). T1 and T2 are at the time points of dust outbreak in DSR, while
the T3 and T4 are at time points of dust pollutants being transported to NCP. All of
the four key time points correspond to peak [PMC] change (**Fig 6a, 6c**). To capture
different dust pollution phases, we analyzed the [PMC] change distributions for these
time points (**Fig. 7**). At T1, the dust storm started and was limited in DSR (**Fig. 7a**).
Hence, ERPs caused prominent [PMC] decrease in DSR (-16.7 μg m$^{-3}$), whereas had
small influence in NCP (lower than 2.0 μg m$^{-3}$ both in NCP and BTH) (**Fig. 6a, Fig.
7b**). At T2, dust storm was transported from DSR to NCP. As a result, the [PMC]
values were diluted in DSR, while were enhanced in NCP (**Fig. 7c**). [PMC] decrease
was considerable in DSR (-8.0 μg m$^{-3}$), and there was a significant [PMC] decrease in
northern NCP by about -10.0 to -30.0 μg m$^{-3}$ (**Fig. 7d**). At T3, the dust storm moved
from the source region to the downwind NCP region (**Fig. 7e**). The ERPs significantly
reduced the dust pollution in the NCP region (**Fig. 7f**), causing the remarkable [PMC]
reduction in BTH (-19.3 μg m$^{-3}$) and NCP (-9.3 μg m$^{-3}$) (**Fig. 6a, 6c**). At T4, it was
the point of the end of the dust episode, and the [PMC] values wee started to decrease
(**Fig. 7g**).
**4 Summary and conclusions**
Dust pollution has significant impacts on human's life in China, especially in
springtime. To reduce dust pollution problem, Chinese government has taken great
efforts for initiating national ecological restoration programs (ERPs) since 1978.
Despite the incontestably great changes of surface properties induced by ERPs, the
effectiveness of ERPs in dust pollution control is not well understood. In the present
study, we are trying to assess the impact of ERPs on the dust pollutions, especially in
the downwind populated region (NCP). First, the ERPs induced land cover changes
are investigated, using the long-term satellite measurements. The gridded LUF
matrixes are calculated and then assimilated, which can provide more accurate surface
properties than previous studies, especially for the dust emissions due to wind erosion
in the WRF-DUST model. Second, the WRF-DUST model is applied to evaluate the
effects of the ERPs on the dust pollution control in NCP. Some important results are
summarized as follows:
1. A more detailed land surface properties are quantified by calculating gridded LUF



based on long-term satellite measurement. Two important vegetation (grass and
forest) are increased in berried lands and deserts in northwestern China, which
locate in the upwind regions of the populated areas of NCP in eastern China. As a
result, China has impressive progress in implementing some of the world's largest
ERPs, which could produce important impacts on the dust pollution in eastern of
China.
2. The WRF-DUST model is applied to assess the effect of ERPs on dust pollutions.
The model calculations are intensively evaluated. Despite some model biases, the
WRF-DUST model reasonably reproduced the temporal and spatial dust pollution
episode both in upwind DSR and downwind NCP regions, especially for the dust
storm outbreak and the down wind transport. The correlation coefficients (R)
between simulated and observed [PMC] are 0.96 for DSR and 0.83 for NCP, and
the NMBs are 2% and -15%, respectively.
3. The impacts of EPRs induced land cover changes on the dust pollution in NCP are
assessed during an episode of dust storm (from 02 to 07 March, 2016). The results
suggest that ERPs significantly reduce the dust pollution in NCP, especially in the
heart area of NCP (BTH). During the episode when the dust storm was transported
from the DSR to NCP, the reduction of dust pollution induced by ERPs ranges
from -5% to -15% in NCP, with the maximum reduction of -15.3% (-21.0 $\mu g\ m^{-3}$)
in BTH, and -6.2% (-9.3 $\mu g\ m^{-3}$) in NCP.
The air pollution is severe in eastern China, especially in NCP, and the dust pollutions
have important contributions to the severe air pollutions. This study shows that ERPs
help to reduce some air pollutions in the region, especially in springtime, suggesting
the important contribution of ERPs to the air pollution control in China. It should be
reiterated that, considering the limitation of case study and the sparse empirical
evidence, the main focus of this study does not intent to give a general conclusion, but
rather to provide some insights of the effect of ERPs on the downwind area, where
heavy haze often occurred due to anthropogenic air pollutants.





**Acknowledgement**
This work is supported by the National Natural Science Foundation of China (NSFC)
under Grant No. 41430424 and 41375136, and the China Postdoctoral Science
Foundation under Grant No. 2016M602886. This work is also supported by the
Fundamental Research Funds for the Central Universities. The National Center for
Atmospheric Research is sponsored by the National Science Foundation.

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



**Figure Captions**


**Figure 1**. WRF-DUST simulation domain with surface land properties and major
natural dust sources in China. The crosses represent centers with ambient
monitoring sites. The land cover properties are derived from the MCD12Q1
product in the year 2013. Distribution of Gobi and deserts are adapted from
1:200,00 desert distribution dataset provide by the Environmental and
Ecological Science Data Center for West China, National Natural Science
Foundation of China (http://westdc.westgis.ac.cn).
**Figure 2**. The horizontal distributions of land cover changes induced by the ERPs
from 2001 to 2013 for the categories of **(a)** barrens, **(b)** grasslands/savannas,
**(c)** forest, and **(d)** others.
**Figure 3**. The temporal variations of predicted (read lines) and observed (black dots)
diurnal profiles of near-surface [PMC] over all ambient monitoring stations
in provinces within regions of DSR and NCP. The model performance
statistics of *NMB*, and *IOA* are also shown. The x-axis is the date in Beijing
Time.
**Figure 4**. The comparison of calculated (color contour) and observed (colored circles)
episode average [PMC]. (**a**) [PMC] distribution along with the simulated
wind fields (black arrows). (**b**) The correlation analysis.
**Figure 5**. The distribution of calculated (color contour) and observed (colored circles)
daily average [PMC], along with the simulated wind fields (black arrows).
The correlation indices (R) between measurements and simulations are also
presented.
**Figure 6**. The impacts of ERPs on near-surface [PMC] in regions of DSR, NCP and
BTH, including (**a, c**) the temporal variations and (**b, d**) the
episode-average spatial variations. Both the concentration (**a, b**) and
percentage (**c, d**) influences are presented.
**Figure 7**. The horizontal distributions of (**a, c, e, g**) [PMC] and (**b, d, f, h**) [PMC]
change for the key time points of T1, T2, T3, and T4 (see Fig. 6). The
pattern comparisons of simulated vs. observed [PMC] are shown in left
panels, as well as their correlation indices, along with the simulated wind
field.
**Figure 1**

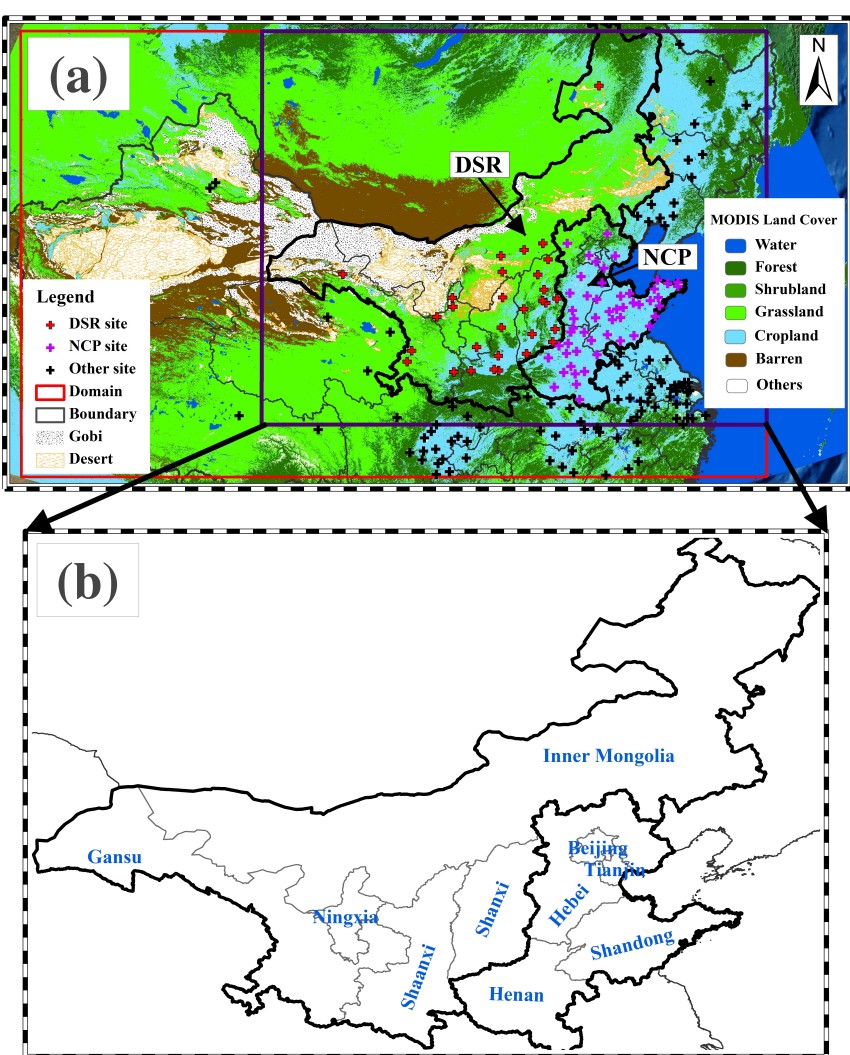


**Figure 1**. **(a)** WRF-DUST simulation domain with surface land properties and major
natural dust sources in China. **(b)** The details of ROIs for the dust source region and
surrounding areas (DSR) and the downwind North China Plain (NCP) region. The
crosses represent centers with ambient monitoring sites. The land cover properties are
derived from the MCD12Q1 product in the year 2013. Distribution of Gobi and
deserts are adapted from 1:200,00 desert distribution dataset provide by the
Environmental and Ecological Science Data Center for West China, National Natural
Science Foundation of China (http://westdc.westgis.ac.cn). The DSR region contains
five provinces in the northwest of NCP, involving Ningxia, Gansu, Shanxi, Inner
Mongolia and Shanxi. The NCP includes five provinces of the Beijing, Tianjin, Hebei,
Henan and Shandong.



**Figure 2**

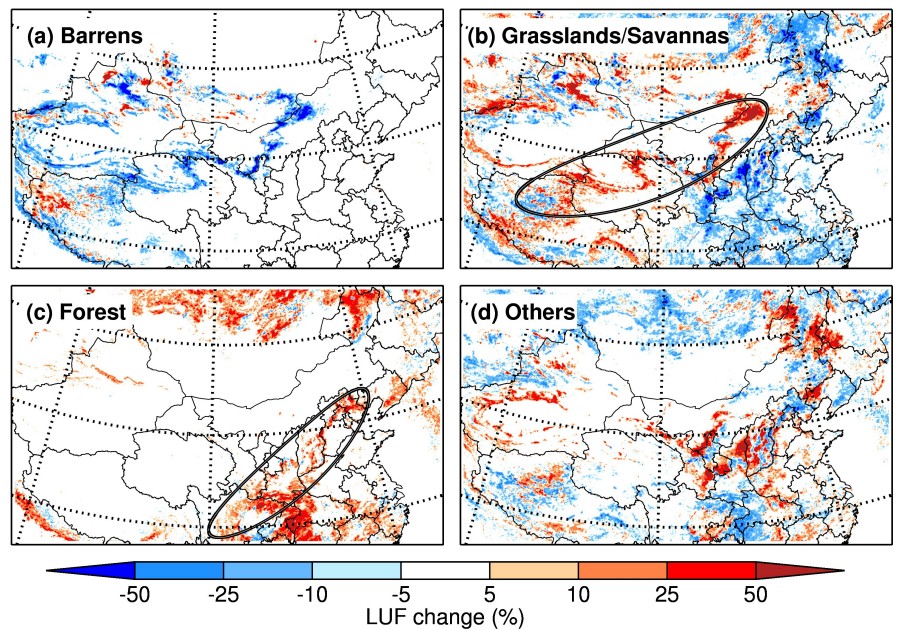


**Figure 2**. The horizontal distributions of land cover changes induced by the ERPs
from 2001 to 2013 for the categories of **(a)** barrens, **(b)** grasslands/savannas, **(c)** forest,
and **(d)** others.





**Figure 3**

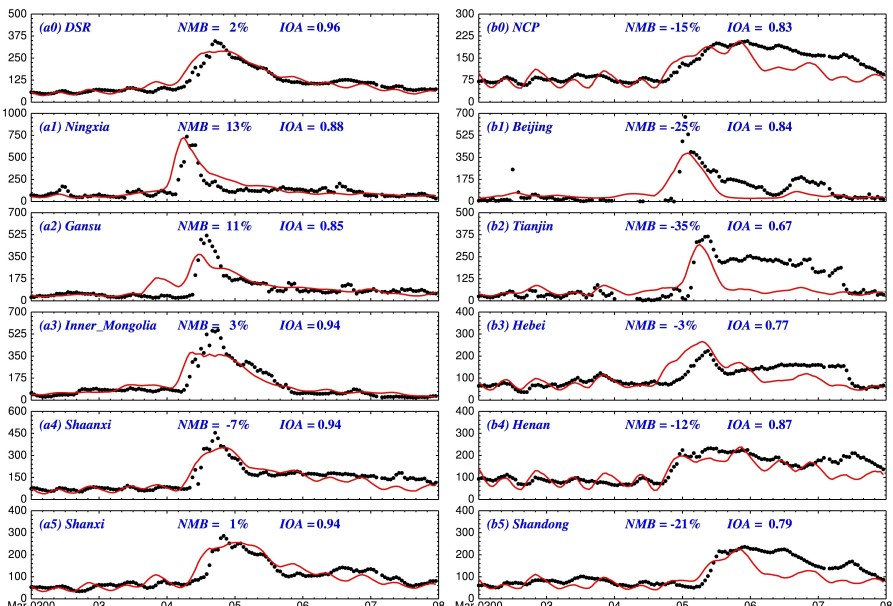


**Figure 3**. The temporal variations of predicted (read lines) and observed (black dots)
diurnal  profiles  of  near-surface  [PMC]  over  all  ambient  monitoring  stations  in
provinces within regions of DSR and NCP. The model performance statistics of *NMB*,
and *IOA* are also shown. The x-axis is the date in Beijing Time.

上





**Figure 4**

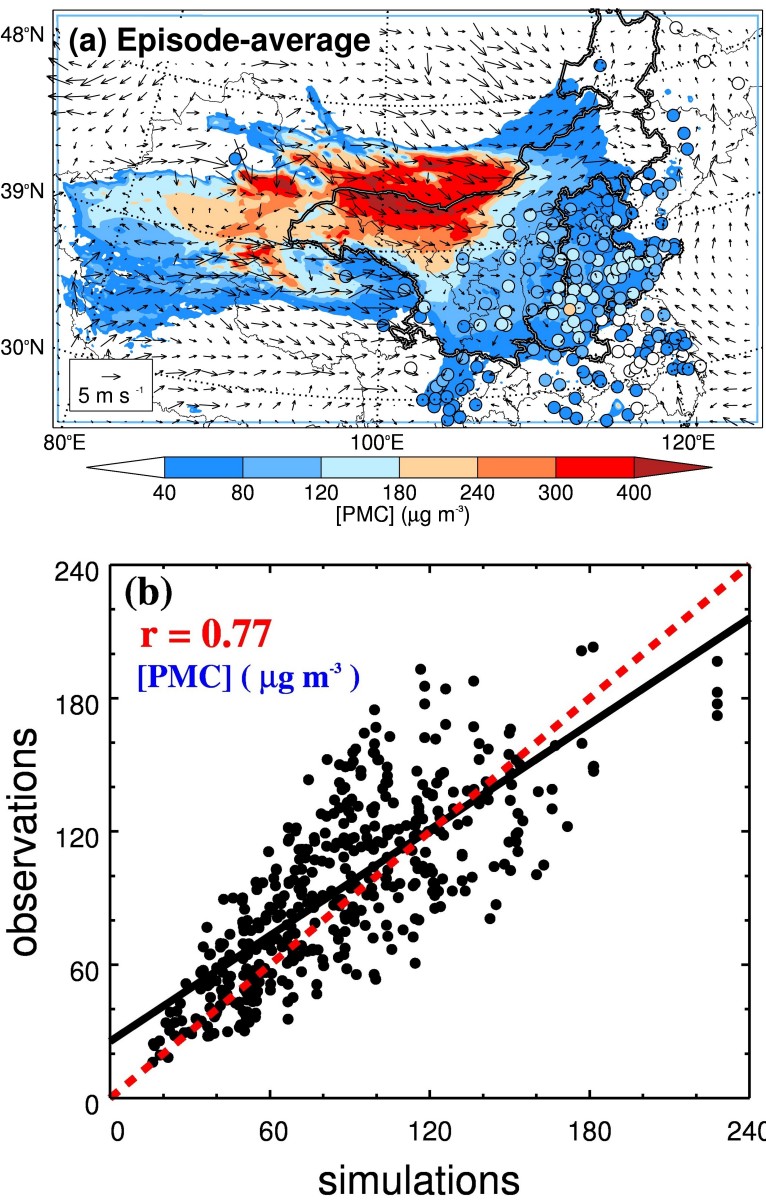


**Figure 4**. The comparison of calculated (color contour) and observed (colored circles)
episode average [PMC]. (**a**) [PMC] distribution along with the simulated wind fields
(black arrows). (**b**) The correlation analysis.





**Figure 5**

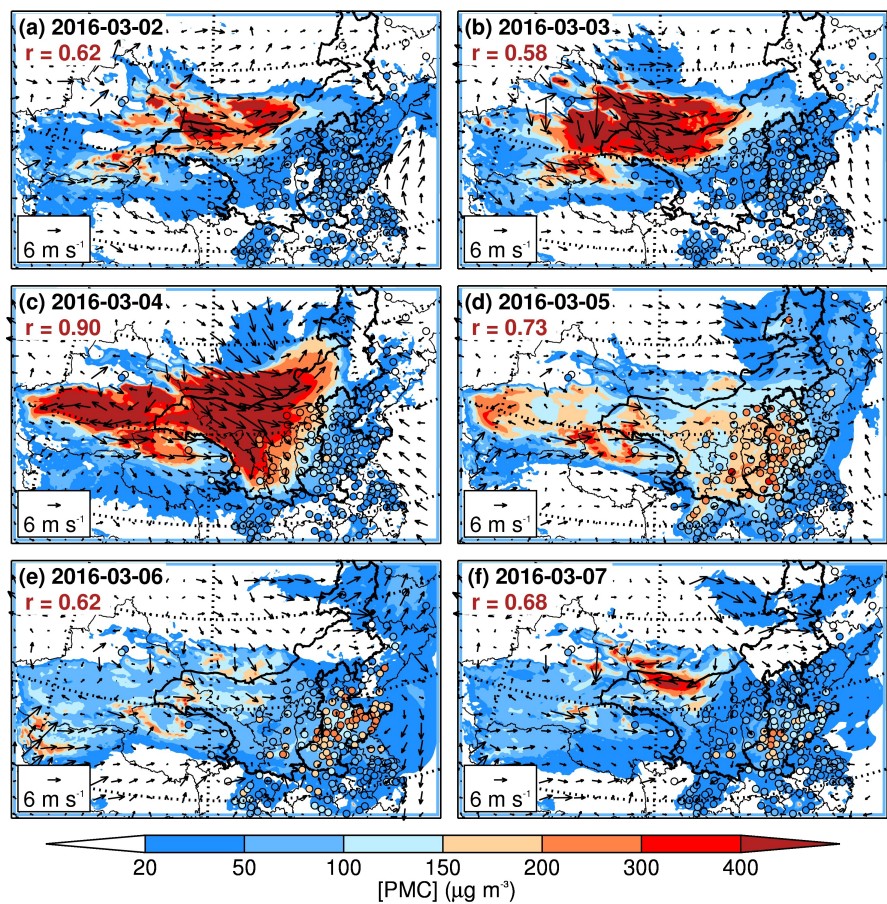


**Figure 5**. The distribution of calculated (color contour) and observed (colored circles)
daily average [PMC], along with the simulated wind fields (black arrows). The
correlation indices (R) between measurements and simulations are also presented.



**Figure 6**

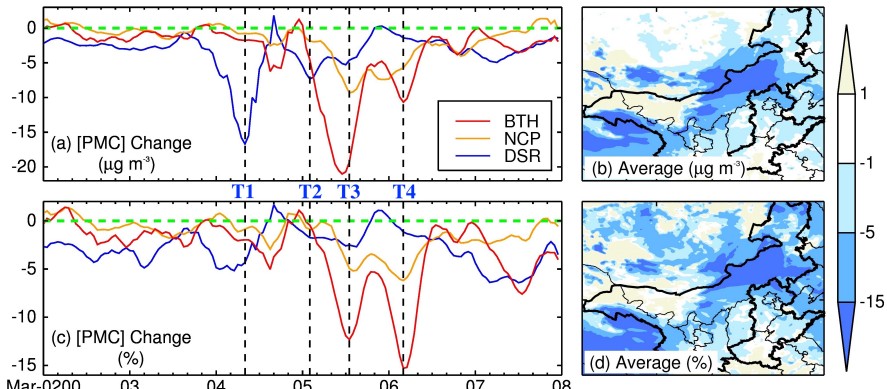


**Figure 6**. The impacts of ERPs on near-surface [PMC] in regions of DSR, NCP and
BTH, including (**a, c**) the temporal variations and (**b, d**) the episode-average spatial
variations. Both the concentration (**a, b**) and percentage (**c, d**) influences are
presented.

**Figure 7**

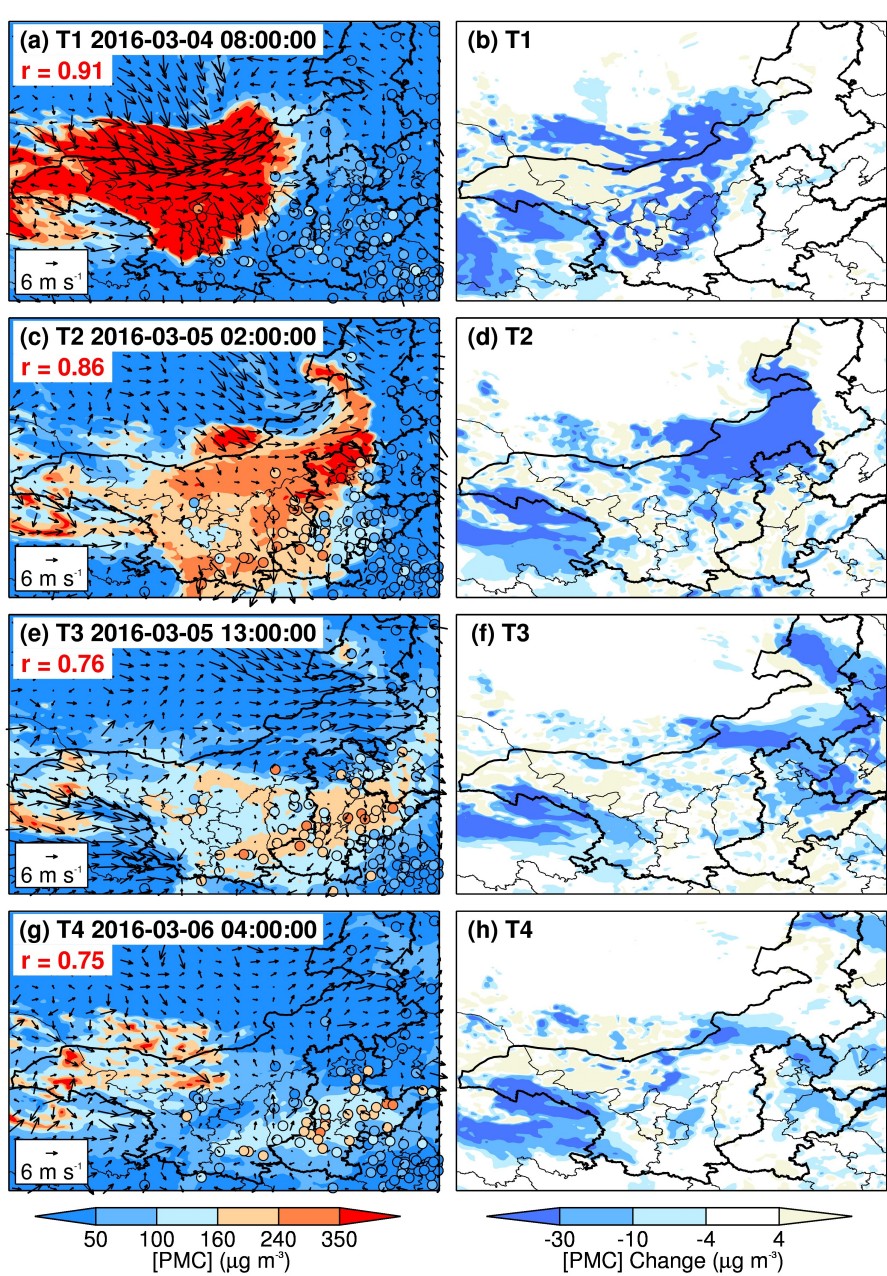


**Figure 7**. The horizontal distributions of (**a, c, e, g**) [PMC] and (**b, d, f, h**) [PMC]
change for the key time points of T1, T2, T3, and T4 (see Fig. 6). The pattern
comparisons of simulated vs. observed [PMC] are shown in left panels, as well as the
correlation indices, along with the simulated wind field.