# Peer review of "Effect of ecological restoration programs on dust pollution in North China Plain: a case study"

_Atmospheric Chemistry and Physics, 2017_

## Referee Comment (RC1) · Anonymous Referee #2 · 2 Dec 2017

Effect of ecological restoration programs on dust polution in North China Plain, China by Xin Long, Xuexi Tie, Guohui Li, Junji Cao, Tian Feng, Li Xeng, and Zhisheng An, submitted for publication in Atmos. Chem. Physics.

The effect of the Ecological Restoration Programs initiated by the Chinese government at reducing air pollution has been analyzed using the regional WRF-DUST model. This is an interesting subject, which is worth publication in Atm. Chem. Phys. A dust episode is first simulated to evaluate the model performances. Then, two experiments are performed each one corresponding to distinct periods of the restoration programs. Their idea is to assess any effect of these programs at reducing suspended dust. Their simulations indicate indeed a sharp decrease of dust, which is quite remarkable. Unfortunately, the authors stops there, and did not explain the physical reasons beyond it.

[Figure]

So, more work is needed here.

An apparent detail but in fact misleading the reader is the tendency of the authors to place dust in the category of pollutants. Dust being produced mechanically by wind erosion (neglecting dust emitted from construction, agriculture, or off-road vehicles) it does not belong to the category of pollutants, which are anthropogenically produced. The manuscript needs a major revision to improve its clarity because poor English. Hopefully my long hours at making suggestions to improve it will help.

In summary, the paper needs major improvements before being publishable but is a potentially interesting paper.

Detailed comments

Line 51: "have wide impacts on the Earth's radiative forcing budget" => "influence the Earth's radiative budget" Line 51: replace "Liao et al., 2005" reference by the more appropriate "Miller and Tegen, 1998"

Line 55: "Distinguished from ...(Moulin et al., 1997)". It depends which period you look at. From 1980 to 2009, there has been a "decreasing dust trend in the tropical North Atlantic is most closely associated with the decrease of Sahel dust emission and increase of precipitation over the tropical North Atlantic, likely driven by the sea surface temperature increase." (Chin et al., 2014). If you look at longer period of Barbados data, you will notice a decrease since 2000.

Line 60: "... and beyond North America to Europe (Grousset el al., 2003)"

Line 61: "There are two dominant source regions of East Asian dust storms locate in China"=> "There are two major sources of dust in China"

Line 63: You may want also to mention dust sources from desertification, agricultural practices (Ginoux et al., 2012), and construction (Long et al., 2016).

Lines 63-65: Remove this sentence as you already mentioned dust influence on air

quality (Line 53).

Line 66: "dust pollution" is not really adequate. Dust is essentially produced mechanically by wind erosion, which have for the most part not been disturbed by human activities. On the other hand, precursors of pollutants are emitted by human activities.

Line 70: When did the "Green Wall of China" started?

Line 74: "However…ERPs." Unclear. Reformulate.

Lines 86-88 should be moved at the beginning of the Introduction, and check for repetition of dust impacts.

Lines 89-90: Unclear and seems unrelated to the present work

Lines 91-92: Unclear, reformulate. Do you mean?: "Few studies have been so far dedicated to estimate the effectiveness of ERPs in controlling dust erosion"

Line 93 "in regional scale" => "on regional scale" Lines 91-103: To help posing the problem more clearly I suggest starting the paragraph with a sentence like "One of the main difficulty in evaluating the effectiveness of ERPSs is to separate vegetation change by ERPs from other factors, including climate change or CO2 fertilization."

Line 105: remove "first-hand sources"

Line 106: "WRF-DUST model" =>"regional WRF-DUST model".

Line 107: "MODIS land cover". You may want to justify your choice by referring to Wu et al. (2008) in section 2.2

Line 110: "We selected two regions.." It took me a while to figure where were these 2 regions in Figure 1. It would greatly help to use two different colors to differentiate them.

Line 120: "has commenced" => "started"

Line 125: "detailed"=> "speciation of"

Line 127: "utilized" =>"used"

Line 129: "model" => "mode"

Line 130: "can effectively decrease the uncertainty of anthropogenic fine particulate matter" => "is an efficient way to avoid contribution from anthropogenic fine mode particles"

Line 133: "research domain" => ROIs

Line 135: "the most measurement sites (. . .) locate" => "most measurements sites (. . .) are located"

Line 137: "provides a good opportunity. . .evolution" => "provides sufficient spatial coverage to follow the evolution of dust plumes"

Line 145: Add a few words about the evaluation of different land cover datasets over China by Wu et al. (2008).

Line 154: "mosaicked" => "mosaic"

Line 156: "We conducted the geospatial processing.."=> "We processed MCD12Q1 data to fit with WRF-CHEM resolution"

Line 158-174: These are too technical, and not helpful for our understanding. Remove or move it too supplemental material. On the other hand, it would be informative to justify the use of a linear relationship between LUF and dust emission relative to previous studies. Some studies have also used linear relationship (e.g. Werner et al., 2003), but other chosen instead an exponential dependency (e.g. Evans et al., 2016), or threshold (Kim et al., 2010).

Line 175: "WRF-DUST model and configuration" => "Model description"

Section 2.3: You should mention if in the model interactions of dust particles with radiation and cloud microphysics are included, and what are the optical properties used.

We need to know if you are using strongly absorbing or scattering dust optical property, as it will affect the hydrological cycle and subsequently dust deposition. More fundamentally, you need to let the reader know if feedbacks are possible but have not been analyzed in the present study.

Section 2.3: It should describe the base case and the experiments (before and during the ERPs)

Line 179: "Chin et al., 2000" => The description of GOCART is by Chin et al. (2002) and dust scheme by "Ginoux et al., 2001"

Lines 195-198: "Because the dust emissions are strongly dependent on different categories of land cover, to better . . .category." => Split into 3 sentences. To help you: "Dust emission depends on surface properties, such as vegetation cover and soil types, such that we include a dependency on land cover in the emission scheme (Eq. 2)."

Line 200 Eq 3: I don't understand this. First E should have an index k. Secondly, what will happen if within one grid box you have multiple land cover types. You should have a sum over all land covers. You should include the values of $E_k$ in a Table.

Line 206: "The WRF-DUST model adopts one grid with horizontal resolution of 9 km"=>"The domain centered at (112E, 41 N) is composed horizontally of 500 by 300 grid points spaced with a resolution of 9 km, and vertically. . ."

Line 213: Reference missing: Kalnay et al. (1996)

Line 214:=> "Each case studies are simulated over X days with 3 days for spin-up."

Line 214: "impacts" Which impact? Line 216: "include in the calculation" What calculation? Line 216: "detailed emission inventory" Inventory of what?

Line 221: "model calculation" => "model results" Line 224: You already provided the reasons for doing such analysis Line 220. Remove this repetition

Line 237: It is not sufficient to use satellite data to assert that ERP is responsible for

lad cover changes. It may be as well due to changes in hydrological cycle. A reference using in-situ with satellite data would be more convincing.

Line 239: "northwester" => "northwestern" Figure 2: "Barrens" is incorrectly used. Change to "Bare soil" or "Bare surface"

Line 249-252: Reformulate this sentence more clearly. Also the increased forest cover is not related to a decrease of bare surface (Fig 2a), which means that it was not emitting dust initially. Therefore the forest acts as a barrier for dry deposition and not emission. You should precise this important point. Furthermore, the forest will impact dust load only if dust plumes evolve in the boundary layer. This is not always the case as they move up along cold fronts.

Line 257: "They share . . ."=> " These programs help at protecting grassland and reducing desertification."

Section 3.2: You need to include some description of the vertical profile over the ERPs. Is the dust plume in contact with the surface or not? It is fundamental to know this if you one to study ERPs effect on dust.

Line 267: Figure 3. It is not possible to locate these sites on a map. I would suggest showing them on Figure 4 replacing black by red color the circles showing the location of all sites.

Line 311: "suggesting . . . period." => "Indicative of a good model skill at simulating the evolution of the dust plume"

Section 3.4: You should go beyond describing the figures. Why is dust decreasing? Is it an increased deposition: wet or dry? What about the emission? Are they the same? Vertical profiles? Are they similar? This section needs to be work out to provide some scientific content to the study.

Line 341: "The evaluation the model"=> "The evaluation of the model" Lines 341-344: You already said that the model performed well. You repeat yourself. Remove.

Lines 345-348: This was already said in the Introduction. You repeat yourself. Remove.

Line 352-356: It would be better to move this descriptive part in the "WRF-DUST model and experiments". Also, shorten this by saying: "We performed two experiments, one in 2001 before the implementation of the ERPs and the other in 2013 corresponding to its mature phase." On the other hand, you need to provide more information about the simulation. Is this a full one-year simulation? What is the spin-up time? Are the initial conditions for the aerosols identical for both experiments?

Line 357: "from 2001 to 2013" means that you did a 13-year simulation. Is this what you did? You did not define the length of simulation for your experiments. And Figure 6 is poorly described. We have no idea what is the X-axis: year? Month? Day? Hour? Something else? We have no idea what is the y-axis? What are the units if any?

Lines 358-360: "The vegetation increase regions and downwind areas..." => "Regions with increased vegetation (cf. Fig 2b) and their downwind areas..." Line 360: "barren" => "bare surfaces"

Lines 405-410. Needs to be reformulated to use proper English

Line 411: "The WRF-DUST...pollutions" remove, as this is not a result.

Line 412: "The model calculations are intensively evaluated." => "The model results have been evaluated by comparing with surface data." But this is not a result and should be moved earlier in the section. Also the results of statistical analysis is crucial for any model, I would not define it as "important" as you are not the first modeler to use WRF-DUST.

Item 4 missing: You should add a physical explanation of the ERPs effect on dust. Is this due to increase deposition (wet or dry?) or emission? Is there any feedback?

Line 425: "dust pollutions" => "dusty episodes" Line 425: Awkward sentence: "The air pollution is severe . . . to the severe air pollutions"

Line 426: "ERPs help reduce some air pollutions". This is misleading. There is a clear difference between air pollution, which refers to aerosol particles produced by oxidation of anthropogenic precursors, and mineral dust particles mostly produced by wind erosion. Previous modeling study by Chin et al. (2014) shows a sharp decrease of pollution from 1990 to 2010 but an overall increase from 1980, while dust is staying pretty much constant over East Asia. You should check the entire manuscript for similar misleading definition of pollution.

Lines 430-432: Awkward sentence. Please reformulate and do not use the word "sketchy" to characterize your work. I don't think that Atm. Chem. Phys. will publish "sketchy" work.

Line 655: "barrens"=>"bare soil" or "bare surface" and change in the Figure

Figure 3: What is the X-axis: day, month, and year? What are the units for the Y-axis? Replace "The model performance statistics o NMB and IOA are also shown" by providing the full name of NMB and IOA (I even cannot find it in the text!).

Figure 4 caption: What is the period covered by "episode average"? Figure 4 b is not a "correlation analysis". Define circles and lines.

Figure 5 caption: "The correlation indices (R) between measurements and simulations are also presented"=> "The correlation coefficient (r) of the linear regression between simulated and observed surface concentration is indicated in red."

Figure 6 caption: Need major improvement, as it is impossible to know what is shown on this Figure from the caption

Figure 7: Right panels are not defined properly as I have no clue what they are. The entire caption needs improvement for clarity.

Chin, M., Ginoux, P., Kinne, S., Torres, O., Holben, B. N., Duncan, B. N., ... & Nakajima, T. (2002), Tropospheric aerosol optical thickness from the GOCART model and comparisons with satellite and Sun photometer measurements, J. Atmos. Sciences,

59(3), 461-483.

Chin, M., Diehl, T., Tan, Q., Prospero, J. M., Kahn, R. A., Remer, L. A., Yu, H., Sayer, A. M., Bian, H., Geogdzhayev, I. V., Holben, B. N., Howell, S. G., Huebert, B. J., Hsu, N. C., Kim, D., Kucsera, T. L., Levy, R. C., Mishchenko, M. I., Pan, X., Quinn, P. K., Schuster, G. L., Streets, D. G., Strode, S. A., Torres, O., and Zhao, X.-P.: Multi-decadal aerosol variations from 1980 to 2009: a perspective from observations and a global model, Atmos. Chem. Phys., 14, 3657-3690, https://doi.org/10.5194/acp-14-3657-2014, 2014.

Evans, S., P. Ginoux, S. Malyshev, and E. Shevliakova (2016), Climate-vegetation interaction and amplification of Australian dust variability, Geophys. Res. Lett., 43, 11,823–11,830, doi:10.1002/2016GL071016.

Ginoux, P., Chin, M., Tegen, I., Prospero, J. M., Holben, B., Dubovik, O., & Lin, S. J. (2001), Sources and distributions of dust aerosols simulated with the GOCART model, J. Geophys. Res.: Atmospheres, 106(D17), 20255-20273.

Ginoux, P., J. M. Prospero, T. E. Gill, N. C. Hsu, and M. Zhao (2012), Global-scale attribution of anthropogenic and natural dust sources and their emission rates based on MODIS Deep Blue aerosol products, Rev. Geophys., 50, RG3005, doi:10.1029/2012RG000388.

Grousset, F., P. Ginoux, A. Bory, and P. Biscaye (2003), Case study of a Chinese dust plume reaching the French Alps, Geophys. Res. Lett., 30, 1277, doi:10.1029/2002GL016833, 6.

Kalnay, E., Kanamitsu, M., Kistler, R., Collins, W., Deaven, D., Gandin, L., ... & Zhu, Y. (1996), The NCEP/NCAR 40-year reanalysis project, Bulletin of the American meteorological Society, 77(3), 437-471.

Kim, D., Chin, M., Kemp, E.M., Tao, Z., Peters-Lidard, C.D. and Ginoux, P. (2017), Development of high-resolution dynamic dust source function-A case study with a strong

dust storm in a regional model, Atmospheric Environment, 159, 11-25.

Long, Xin, Nan Li, Xuexi Tie, Junji Cao, Shuyu Zhao, Rujin Huang, Mudan Zhao, Guohui Li, and Tian Feng: Urban dust in the Guanzhong Basin of China, part I: A regional distribution of dust sources retrieved using satellite data, Science of the Total Environment 541 (2016): 1603-1613.

Miller, R. L., and I. Tegen, 1998: Climate response to soil dust aerosols. J. Climate, 11, 3247–3267, doi:https://doi.org/10.1175/1520-0442(1998)011

Werner, M., I. Tegen, S. P. Harrison, K. E. Kohfeld, I. C. Prentice, Y. Balkanski, H. Rodhe, and C. Roelandt, Seasonal and interannual variability of the mineral dust cycle under present and glacial climate conditions (2002), J. Geophys. Res., 107(D24), 4744, doi:10.1029/2002JD002365.

Wu, W., Shibasaki, R., Yang, P., Ongaro, L., Zhou, Q. and Tang, H., 2008: Validation and comparison of 1 km global land cover products in China. International Journal of Remote Sensing, 29(13), 3769-3785. doi:https://doi.org/10.1080/01431160701881897
* * *

---

## Referee Comment (RC2) · Anonymous Referee #1 · 5 Jan 2018

This manuscript provides a case study of changed landuse fraction on the dust storm over Northern China. Its method is straightforward and easy to understand. My main concern is whether the single case study of 5 days is sufficient for the climatological pattern shift of dust storm as the paper title states. The authors may consider study more cases for more years. For instance, this single case shows that the dust storms strength became weaker after changing its landuse. How about the frequency of the dust storm occurrence for one or more year? It would be better to add more convincing cases.

---

## Author Comment (AC3) · 31 Jan 2018

**Response to Referee #2**

We thank the reviewers for the careful reading of the manuscript and helpful comments. According to the suggestions of the reviewer, the reviewers' comments have been carefully addressed, and the paper is carefully revised. We believe that the revised paper has been significantly improved after addressing the comments of the reviewers.

\*\*\*\*\*\*\*\*\*\*\*\*\*\*\*\*\*\*\*\*\*\*\*\*\*\*\*\*\*\*\*\*\*\*\*\*\*\*\*\*\*\*\*\*\*\*\*\*\*\*\*\*\*\*\*\*\*\*\*\*\*\*\*\*\*\*\*\*\*\*\*\*\*\*\*\*\*\*

**General comments:**
* * *
**The effect of the Ecological Restoration Programs initiated by the Chinese government at reducing air pollution has been analyzed using the regional WRF-DUST model. This is an interesting subject, which is worth publication in Atm. Chem. Phys. A dust episode is first simulated to evaluate the model performances. Then, two experiments are performed each one corresponding to distinct periods of the restoration programs. Their idea is to assess any effect of these programs at reducing suspended dust. Their simulations indicate indeed a sharp decrease of dust, which is quite remarkable. Unfortunately, the authors stops there, and did not explain the physical reasons beyond it. So, more work is needed here.**

To response to reviewer's comments, we add more analyses in the revised version, including:

(1) More model simulations are included in the revised version. Additional three sensitivity experiments are conducted to enhance the model results in the revised manuscript (see Fig. S3, S4, and S5).

(2) More analyses are added to explain the physical reasons of the reduction of dust concentration due to the Ecological Restoration Programs. The summary of analysis is shown in the revised abstract. In addition, to clearly explain the physical reason, we add a new figure to show the vertical profiles to

investigate the reasons of dust decrease (Fig. 9).

(3) We add a new section to explain in details of the physical reason (see Section 3.4 Reasons for dust decreasing).
* * *
**An apparent detail but in fact misleading the reader is the tendency of the authors to place dust in the category of pollutants. Dust being produced mechanically by wind erosion (neglecting dust emitted from construction, agriculture, or off-road vehicles) it does not belong to the category of pollutants, which are anthropogenically produced. The manuscript needs a major revision to improve its clarity because poor English. Hopefully my long hours at making suggestions to improve it will help.**

**In summary, the paper needs major improvements before being publishable but is a potentially interesting paper.**

To address the reviewer's comments, we change the "*dust pollutants*" to "dust particles" in the entire text. And the "dust pollutions" was also replaced by "dust concentration" or "dust plumes".

We thank for the hard works of the reviewer to correct the English of the paper, and all the English corrections are included in the revised manuscript. With this reviewer's help, the English in the revised version is significantly improved.

**Detailed comments**
* * *
**1.** **Line 51: "have wide impacts on the Earth's radiative forcing budget" =>
"influence the Earth's radiative budget" Line 51: replace "Liao et al., 2005"
reference by the more appropriate "Miller and Tegen, 1998"**

>   **In Line 50,** we revised the text: "influence the Earth's radiative budget". And we
>   also replaced the reference of "Liao et al., 2015" by "Miller and Tegen, 1998".
* * *
**2.** **Line 55: "Distinguished from ...(Moulin et al., 1997)". It depends which
period you look at. From 1980 to 2009, there has been a "decreasing dust trend
in the tropical North Atlantic is most closely associated with the decrease of
Sahel dust emission and increase of precipitation over the tropical North Atlantic,
likely driven by the sea surface temperature increase." (Chin et al., 2014). If you
look at longer period of Barbados data, you will notice a decrease since 2000.**

>   **In Lines 57,** to address the comment of the reviewer, we delete the inaccurate
>   description of this sentence, and add more text as descript by the reviewer: "From
>   1980 to 2009, there has been a decreasing dust trend in the tropical North Atlantic,
>   being most closely associated with the decrease of Sahel dust emission and
>   increase of precipitation over the tropical North Atlantic, and likely driven by the
>   sea surface temperature increase (Chin et al., 2014)."
* * *
**3.** **Line 60: " … and beyond North America to Europe (Grousset el al., 2003)"**

>   **In Line 65,** we added this point and updated the reference: "… and beyond North
>   America to Europe (Grousset et al., 2003)."
* * *
**4.** **Line 61: "There are two dominant source regions of East Asian dust storms
locate in China"=> "There are two major sources of dust in China".**

**In Line 66,** we revised the description: "There are two major sources of dust in China..."
* * *
**5.   Line 63: You may want also to mention dust sources from desertification, agricultural practices (Ginoux et al., 2012), and construction (Long et al., 2016).**

**In Line 68,** we revised the dust sources description: "Dust particles come from many different sources, such as desertification, agricultural practices (Ginoux et al., 2012), construction (Li et al., 2016), and regional transport from exposed lands (Bian et al, 2012; Su et al., 2017)."
* * *
**6.   Lines 63-65: Remove this sentence as you already mentioned dust influence on air quality (Line 53).**

We removed the sentence.
* * *
**7.   Line 66: "dust pollution" is not really adequate. Dust is essentially produced mechanically by wind erosion, which have for the most part not been disturbed by human activities. On the other hand, precursors of pollutants are emitted by human activities.**

In the revised manuscript, the "*dust pollution*" is changed to "dust particles" or "dust plumes".
* * *
**8.   Line 70: When did the "Green Wall of China" started?**

It should be "the Green Great Wall of China". China launched its "Green Great Wall" (GGW) program in 1978 (*Fang et al., 2001*).
**In Line 77,** we revised the mistake and add a reference: "As a result, the "Green Great Wall" of China has been established in North China (Fang et al., 2001; Duan et al., 2011)."
* * *
**9. Line 74: "However…ERPs." Unclear. Reformulate.**

**In Line 82,** we revised the text: "However, there is an ongoing debate about the effectiveness of national ERPs."
* * *
**10. Lines 86-88 should be moved at the beginning of the Introduction, and check for repetition of dust impacts.**

**In Line 54,** we moved the description to the beginning of the Introduction: "The mineral dust particles can also serve as carriers and reaction platforms, and the heterogeneous dust chemistry may change the photochemistry, acid deposition, and production of secondary aerosols in the atmosphere (Lou et al., 2014; Fu, 2016; Zhou et al., 2016)."
* * *
**11. Lines 89-90: Unclear and seems unrelated to the present work.**

We removed the descriptions
* * *
**12. Lines 91-92: Unclear, reformulate. Do you mean?: "Few studies have been so far dedicated to estimate the effectiveness of ERPs in controlling dust erosion".**

**In Line 95,** to express more clearly, we revised the descriptions: "Few studies have been so far dedicated to estimate the effectiveness of ERPs in controlling dust erosion on regional scale."
* * *
**13. Line 93 "in regional scale" => "on regional scale" Lines 91-103: To help posing the problem more clearly I suggest starting the paragraph with a sentence like "One of the main difficulty in evaluating the effectiveness of ERPSs is to separate vegetation change by ERPs from other factors, including climate change or CO2 fertilization."**

**In Line 96,** We replaced "in regional scale" by "on regional scale".

We rewrote some parts of the paragraph:

**In Line 95,** "Few studies have been so far dedicated to estimate the effectiveness of ERPs in controlling dust erosion on regional scale."

**In Line 100,** "On the other hand, it is difficult to separate vegetation change by ERPs from other factors, including climate change or $CO_2$ fertilization (Silva et al., 2013). The climate factors are asserted to be one of the main causes for the observed decrease of dust storms in northern China (Cao et al., 2011)."

**In Line 108,** "The previous studies didn't quantify the roles of ERPs on dust concentrations, such as the detailed land cover change induced by ERPs, the effect of regional dust transport to downwind regions, especially in the NCP region, etc."
* * *
**14. Line 105: remove "first-hand sources"**

We deleted the text of "first-hand sources of".
* * *
**15. Line 106: "WRF-DUST model" =>"regional WRF-DUST model".**

**In Line 116,** we revised the text: "regional WRF-DUST model".
* * *
**16. Line 107: "MODIS land cover". You may want to justify your choice by referring to Wu et al. (2008) in section 2.2**

We mentioned the work of Wu et al., (2008) here and give more description of the evaluation by Wu et al., (2008) in other part.

**In Line 112,** "Because the MODIS land cover dataset is a good representative across China (Wu et al., 2008), we …"

**In Line 153,** we added description of the evaluation by Wu et al., (2008): "Wu et al., (2008) compared four global land cover dataset across China, concluding that

the MODIS land cover product is the most representative over China with the minimal bias from the China's National Land Cover Dataset. The MCD12Q1 (Version 5.1)…"
* * *
**17. Line 110: "We selected two regions..." It took me a while to figure where were these 2 regions in Figure 1. It would greatly help to use two different colors to differentiate them.**

The two ROIs of DSR (Dust Source Region) and NCP (North China Plain) were highlighted with different colors **in Figure 1**.
* * *
**18. Line 120: "has commenced" => "started"**
**Line 125: "detailed"=> "speciation of"**
**Line 127: "utilized" =>"used"**
**Line 129: "model" => "mode"**
**Line 130: "can effectively decrease the uncertainty of anthropogenic fine particulate matter" => "is an efficient way to avoid contribution from anthropogenic fine mode particles"**

**In Line 127,** we revised the text: "started"

**In Line 132,** we revised the text: "speciation of"

**In Line 134,** we revised the text: "used"

**In Line 136,** we revised the text: "mode"

**In Line 137,** we revised the text: "… is an efficient way to avoid contribution from anthropogenic fine particles…"
* * *
**19. Line 133: "research domain" => ROIs**

The research domain is settled by the WPS module, and annotated used red line **in Fig. 1a**. The observed cities included the entire black cross marks **in Fig. 1a**. To express it more clearly, we modified **Figure 1** and revised the text:

**In Line 139,** we revised the text: "A total of 184 cities (489 measurement sites) had [PMC] observations in the research domain (see black cross in Fig. 1a), including 30 cities within the DSR region (see magenta cross in Fig. 1b) and 53 cities within the NCP region (see red cross in Fig. 1b)."
* * *
**20. Line 135: "the most measurement sites (…) locate" => "most measurements sites (. . .) are located"**

**Line 137: "provides a good opportunity…evolution" => "provides sufficient spatial coverage to follow the evolution of dust plumes"**

**In Line 142,** we revised the text: "Because the prevailing winds were dominated by west winds, most measurement sites (as shown in Fig. 1a) are located…"

**In Line 144,** we revised the text: "As a result, the China MEP measurement network provides sufficient spatial coverage to follow the evolution of dust concentrations."
* * *
**21. Line 145: Add a few words about the evaluation of different land cover datasets over China by Wu et al. (2008).**

**In Line 153,** we added description of the evaluation by Wu et al., (2008): "Wu et al., (2008) compared four global land cover dataset across China, concluding that the MODIS land cover product is the most representative over China with the minimal bias from the China's National Land Cover Dataset. The MCD12Q1 (Version 5.1)…"
* * *
**22. Line 154: "mosaicked" => "mosaic"**

**Line 156: "We conducted the geospatial processing..."=> "We processed MCD12Q1 data to fit with WRF-CHEM resolution"**

**In Line 164,** we revised the text: "mosaic".

**In Line 165,** we revised the text: "We processed MCD12Q1 data to fit with

WRF-DUST resolution. The gridded land use fraction (LUF) was calculated by Eq. (1)."

**In Line 143,** we revised the word "**adaption**"
* * *
**23. Line 158-174: These are too technical, and not helpful for our understanding. Remove or move it too supplemental material. On the other hand, it would be informative to justify the use of a linear relationship between LUF and dust emission relative to previous studies. Some studies have also used linear relationship (e.g. Werner et al., 2003), but other chosen instead an exponential dependency (e.g. Evans et al., 2016), or threshold (Kim et al., 2010).**

We removed the technical description, and described the dust emission calculation more clearly.

**In Line 195,** we revised the text about the dust emission flux: "The standard version of WRF-CHEM calculates dust emission (Eq. 2) only considering the dominant land cover. The water land cover category can be treated as dominant category only with high LUF greater than 0.5. For other categories (Table S1), the dominant land cover category denotes to the specify one with maximum LUF (Eq. 1) among all the land cover categories excluding water category. Theoretically, one land cover category (excluding water), with LUF greater than the average value (0.05) could be the dominant land cover category. This caused the dust emission calculation in the stand version being insensitive to land cover change, especially for incomplete changes within one grid cell. To better investigate the impacts of ERPs on dust emission, we modified the GOCART dust emission scheme, using the LUF to represent the real dust emission potential. The dust emission flux (G) in each grid is given by the sum of dust emitted from each dust source (Eq. 3).

$$G_p = \begin{cases} \sum_{source} LUF_{source} C\gamma_p E_{source} V^2 (V - V_p) & V > Vt_p \\ 0 & V \le Vt_p \end{cases} \qquad (3)$$

$LUF_{source}$ denotes the gridded area fraction of bare soil and cropland, which are derived from the satellite data (MCD12Q1). The other parameters are the same as

those in Eq. (2). Besides bare soil, we also calculated the largest 'anthropogenic' dust source emitted from agricultural soil (Tegen et al, 2004; Ginoux et al., 2012). We empirically set the erosion factor E=0.12 for cropland and E=0.5 for bare soil in western China (Li et al., 2016a)."
* * *
**24. Line 175: "WRF-DUST model and configuration" => "Model description"**

In Line 173, we revised the text: "**Model description**".
* * *
**25. Section 2.3: You should mention if in the model interactions of dust particles with radiation and cloud microphysics are included, and what are the optical properties used. We need to know if you are using strongly absorbing or scattering dust optical property, as it will affect the hydrological cycle and subsequently dust deposition. More fundamentally, you need to let the reader know if feedbacks are possible but have not been analyzed in the present study.**

In Line 220, we added more information in the model: "…the Goddard long wave radiation parameterization (Chou and Suarez, 1999), and the shortwave radiation parameterization (Chou et al., 2001). The cloud effects to the optical depth in radiation are possible but have not been analyzed in the present study."
* * *
**26. Section 2.3: It should describe the base case and the experiments (before and during the ERPs).**

In Line 285, we added the description of base case (REF case): "We have first adapted the MCD12Q1 product of 2013 into the WRF-DUST model and performed the numerical simulation of dust storm episodes from 2 to 8 March 2016. For the discussion convenience, we have defined the simulation with the 2013 land cover as the reference case (hereafter referred to as REF case), and results from the reference simulation are compared to observations in regions of DSR and NCP."

**In Line 367,** we added the description of ERP related sensitivity case (SEN-ERP case): "As the human activities, especially the national ERPs, are the dominant factor for land cover changes (see Sect. 3.1), we treated the land cover changes related two GGWs as the results of ERPs implementations in the present study. In order to evaluate the impact of the ERPs induced land cover change and resultant [PMC] change, a sensitivity experiment is designed, in which the land cover change related to GGWs (both the grass GGW and forest GGW) derived from MCD12Q1 product of 2001 (see Fig. 2 and Fig. S3) is adapted into the WRF-DUST model, representing the land cover situations without ERPs (hereafter referred to as SEN-ERPs case)."

**In Line 415,** we added the description of other sensitivity cases (SEN-2001, SEN-GRASS and SEN-TREE): "In order to find the presumed reasons for dust decreasing resulted from ERPs induced land cover changes from 2001 to 2013, three additional sensitivity experiments are conducted and compared to REF case. One with the entire land cover changes from 2001 to 2013 (see Fig. 2) adapted into the WRF-DUST model (SEN-2001). Another two experiments are one with the land cover change related to grass GGW (SEN-GRASS case) and the other related to forest GGW (SEN-TREE case). In the SEN-GRASS case, the land cover changes only related to grass GGW is adapted into WRF-DUST model (Fig. S4). It is the same in the SEN-TREE experiment, but adapted with land cover changes only related the forest GGW (Fig. S5)."
* * *
**27. Line 179: "Chin et al., 2000" => The description of GOCART is by Chin et al. (2002) and dust scheme by "Ginoux et al., 2001".**

**In Line 175,** we revised the text and updated the reference: "The GOCART (Georgia Tech/Goddard Global Ozone Chemistry Aerosol Radiation and Transport model) dust scheme (Ginoux et al., 2001; Chin et al., 2002)…"

**In Line 182,** we added the reference of "Ginoux et al., 2001" to the dust emission calculations.
* * *
**28. Lines 195-198: "Because the dust emissions are strongly dependent on different categories of land cover, to better . . .category." => Split into 3 sentences. To help you: "Dust emission depends on surface properties, such as vegetation cover and soil types, such that we include a dependency on land cover in the emission scheme (Eq. 2)."**

We described the dust emission calculation more clearly (see "***Response-23***").
* * *
**29. Line 200 Eq. 3: I don't understand this. First E should have an index k. Secondly, what will happen if within one grid box you have multiple land cover types. You should have a sum over all land covers. You should include the values of E_k in a Table.**

Yes, *E* is a parameter related to surface land cover properties. In the present study, only land surface with bare soil and cropland is considered as possible dust sources. More detailed description can be seen in "***Response-23***"

**In Line 207,** to express clearly, we revised the Eq. 3.

**In Line 212,** we revised the text: "We empirically set the erosion factor E=0.12 for cropland and E=0.5 for bare soil in western China (Li et al., 2016a)."
* * *
**30. Line 206: "The WRF-DUST model adopts one grid with horizontal resolution of 9 km"=>"The domain centered at (112E, 41 N) is composed horizontally of 500 by 300 grid points spaced with a resolution of 9 km, and vertically…"**

**In Line 215,** we revised the text: "The domain centered at (112°E, 41°N) is composed horizontally of 500 by 300 grid points spaced with a resolution of 9 km (**Fig. 1a**) and vertically with 35 sigma levels."
* * *
**31. Line 213: Reference missing: Kalnay et al. (1996)**

**In Line 225,** we added the reference of "Kalnay et al., 1996"
* * *
**32. Line 214: => "Each case studies are simulated over X days with 3 days for spin-up."**

**In Line 226,** we revised the text: "For the episode simulations, the case studies were simulated with 3 days for spin-up."
* * *
**33. Line 214: "impacts" Which impact? Line 216: "include in the calculation" What calculation? Line 216: "detailed emission inventory" Inventory of what?**

**In Line 226,** in order to express clearly, we revised the text: "Considering the contribution of the anthropogenic emission to near surface [PMC], we also calculated the physical process (such as emissions, transport, dry deposition, and gravitational settling) of anthropogenic PMC (coarse mode particle matter) emission in the WRF-DUST simulation. The detailed emission inventory of anthropogenic PMC…"
* * *
**34. Line 221: "model calculation" => "model results" Line 224: You already provided the reasons for doing such analysis Line 220. Remove this repetition**

**In Line 234,** the Sect. 2.4 was revised and merged to Sect. 2.3: "We use the normalized mean bias (*NMB*), the index of agreement (*IOA*), and the correlation coefficient (*R*) to assess the WRF-CHEM model performance in simulating [PMC] against measurements…"

**35. Line 237: It is not sufficient to use satellite data to assert that ERP is responsible for lad cover changes. It may be as well due to changes in hydrological cycle. A reference using in-situ with satellite data would be more convincing.**

The vegetation growth is the most obvious land cover changes, and it is well known as the "Great Green Wall (GGW)" in China. We investigated the distribution of GGWs, and found the human activity (ERPs) is the dominant factor incurred the land cover changes.

**In Line 245,** we deleted the text "*due to the ecological restoration programs (ERPs)*"

**In Line 266,** we added representative examples of reporting for GGWs in the news media: "Governments have claimed a significant contribution of GGWs to control desertification and dust storms, and it has been widely reported in the news media in China (Fig. S2)."

**In Line 268,** we investigated the dependency of GGWs distribution on human activity (ERPs): "The grass GGW acts as barrier to stop the desert move toward to the densely populated area (see Fig. S1c, Fig. S2a, S2b). The forest GGW acts as another barrier to protect the densely populated regions from dust source regions (Fig. S1c)."

**In Line 271,** we added the analysis to explain that human activities, especially ERPs, are the dominant factor for GGWs formation: "The GGWs separate the dust source regions from densely populated and economically developed regions in southeastern China (Fig. S1c), illustrating that the human activities, especially the national ERPs, are the dominant factor for land cover changes, rather than other natural factors, e.g. the natural hydrological cycle."
* * *
**36. Line 239: "northwester" => "northwestern" Figure 2: "Barrens" is incorrectly used. Change to "Bare soil" or "Bare surface"**

**In Line 251,** we revised the text: "…northwestern…"

**In Line 254,** we replaced "barrens" by "bare soil"
* * *
**37. Line 249-252: Reformulate this sentence more clearly. Also the increased forest cover is not related to a decrease of bare surface (Fig 2a), which means that it was not emitting dust initially. Therefore the forest acts as a barrier for dry deposition and not emission. You should precise this important point. Furthermore, the forest will impact dust load only if dust plumes evolve in the boundary layer. This is not always the case as they move up along cold fronts.**

We agree with the reviewer that the two GGWs should be explained more clearly. But we emphasized here the ERPs being the dominant factor of land cover change, and the most obvious land cover changes is the arising of GGWs, mainly resulted from ERPs.

We also added the different between forest barrier and grass barrier **in Sect. 3.4 Reasons for dust decreasing (see *Response-51*)**.

**In Line 460,** we added the analysis about forest barrier.

**In Line 467,** we added the analysis about grass barrier.
* * *
**38. Line 257: "They share…"=> " These programs help at protecting grassland and reducing desertification."**

**In Lines 277,** we revised the text: "These programs help at protecting grassland and reducing desertification."
* * *
**39. Section 3.2: You need to include some description of the vertical profile over the ERPs. Is the dust plume in contact with the surface or not? It is fundamental to know this if you one to study ERPs effect on dust.**

In Line 433, we added the analysis of the vertical profiles of daily [PMC] and [PMC] change for SEN-ERP case and REF – SEN-ERP case (Fig. 9) in Sect. 3.4 Reasons for dust decreasing, and more detailed description can be seen in *Response-51*.
* * *
**40. Line 267: Figure 3. It is not possible to locate these sites on a map. I would suggest showing them on Figure 4 replacing black by red color the circles showing the location of all sites.**

We located the sites of Figure 3 in Fig. 1b, and revised the caption of Figure 3.
* * *
**41. Line 311: "suggesting . . . period." => "Indicative of a good model skill at simulating the evolution of the dust plume"**

In Line 335, we revised the text: "… indicative of a good model skill at simulating the evolution of the dust plumes."
* * *
**42. Section 3.4: You should go beyond describing the figures. Why is dust decreasing? Is it an increased deposition: wet or dry? What about the emission? Are they the same? Vertical profiles? Are they similar? This section needs to be work out to provide some scientific content to the study.**

We conducted other three sensitivity experiments and added Sect. 3.4 Reasons for dust decreasing In Line 414.

The detailed description can be seen in *Response-39* and *Response-51*.
* * *
**43. Line 341: "The evaluation the model"=> "The evaluation of the model"**

**Lines 341-344: You already said that the model performed well. You repeat yourself. Remove.**

We removed the text.
* * *
**44. Lines 345-348: This was already said in the Introduction. You repeat yourself. Remove.**

We removed the text.
* * *
**45. Line 352-356: It would be better to move this descriptive part in the "WRF-DUST model and experiments". Also, shorten this by saying: "We performed two experiments, one in 2001 before the implementation of the ERPs and the other in 2013 corresponding to its mature phase." On the other hand, you need to provide more information about the simulation. Is this a full one-year simulation? What is the spin-up time? Are the initial conditions for the aerosols identical for both experiments?**

Here we performed a case study during a dust storm episode from 2 to 8 March 2016. We added more description for the base case (**In Line 285**) and sensitivity cases (**In Line 367** and **In Line 415**). The detailed information can be seen in *Response-26*.

We also removed the previous unclear description.
* * *
**46. Line 357: "from 2001 to 2013" means that you did a 13-year simulation. Is this what you did? You did not define the length of simulation for your experiments.**

**And Figure 6 is poorly described. We have no idea what is the X-axis: year? Month? Day? Hour? Something else? We have no idea what is the y-axis? What are the units if any?**

We added more description for the base case (**In Line 285**) and sensitivity cases (**In Line 367** and **In Line 415**). The detailed information can be seen in *Response-26*.

To be more clearly, we dived **Figure 6** into two Figures. One is for spatial variations (**Fig.6**) and the other is for temporal variations (**Fig. 7**).

**In Line 375,** we revised the figure description in text: "Figure 6 shows the episode-average near-surface [PMC] change resulted from land cover changes induced by ERPs from 2001 to 2013, including the spatial variations in concentrations (Fig. 6a) and percentage (Fig. 6b) during the episode."

**In Line 388,** we revised the text axis of **Figure 7** and the related description in text: "**Figure 7** presents the hourly near-surface [PMC] change resulted from the land cover changes induced by ERPs from 2001 to 2013, including the temporal variations in concentrations (Fig. 7a) and percentage (Fig. 7b) averaged at monitoring sites in the regions of DSR, NCP and BTH."

We also revised the captions of Fig. 6 and Fig. 7.
* * *
**47. Lines 358-360: "The vegetation increase regions and downwind areas. . ." => "Regions with increased vegetation (cf. Fig 2b) and their downwind areas. . ." Line 360: "barren" => "bare surfaces"**

**In Line 377,** we revised the text: "Regions with increased vegetation (see Fig. 2b, Fig. S2c) and their downwind areas…"

**In Line 380,** we replaced the "*barren*" by "bare surfaces"
* * *
**48. Lines 405-410. Needs to be reformulated to use proper English**

**In Line 494,** we revised the text: "The ERPs resulted in obvious vegetation increase, arising the grass GGW and forest GGW in northwestern China. The GGWs locate between the dust source regions (DSR) and the dense populated North China Plain (NCP) region."
* * *
**49. Line 411: "The WRF-DUST. . .pollutions" remove, as this is not a result.**

We removed the text: "*The WRF-DUST...*"

**In Line 501,** we revised the text: "…temporal and spatial variations of [PMC] during the dust storm episode in both upwind DSR and downwind NCP regions…"
* * *
**50. Line 412: "The model calculations are intensively evaluated." => "The model results have been evaluated by comparing with surface data." But this is not a result and should be moved earlier in the section. Also the results of statistical analysis is crucial for any model, I would not define it as "important" as you are not the first modeler to use WRF-DUST.**

We removed the text.
* * *
**51. Item 4 missing: You should add a physical explanation of the ERPs effect on dust. Is this due to increase deposition (wet or dry?) or emission? Is there any feedback?**

We found the impacts of ERPs on dust concentrations are mainly due to reduction of emission. Other three sensitivity experiments of SEN-2001, SEN-GRASS and SEN-TREE were conducted and compared to REF case. We also investigated the evolution of vertical profiles, analyzing [PMC] (SEN-ERP) and [PMC] change (REF – SEN-ERP) (Fig. 9). At last, we explained the

comparison result (Table 1), combining the land cover change characters (Fig. 2, Fig. S3, Fig. S4, Fig. S5).

**In Line 414,** we added the Sect. **3.4 Reasons for dust decreasing**: "In order to find the presumed reasons for dust decreasing resulted from ERPs induced land cover changes from 2001 to 2013, three additional sensitivity experiments are conducted and compared to REF case. One with the entire land cover changes from 2001 to 2013 (see Fig. 2) adapted into the WRF-DUST model (SEN-2001). Another two experiments are one with the land cover change related to grass GGW (SEN-GRASS case) and the other related to forest GGW (SEN-TREE case). In the SEN-GRASS case, the land cover changes only related to grass GGW is adapted into WRF-DUST model (Fig. S4). It is the same in the SEN-TREE experiment, but adapted with land cover changes only related the forest GGW (Fig. S5).

All the sensitivity experiments are compared to REF case. Table 1 shows the near-surface [PMC] change averaged at monitoring sites in the regions of DSR, NCP and BTH. We found that the forest GGW seems non-significant in dust concentration control with the [PMC] change within 1% in SEN-TREE case. Conversely, the [PMC] changes are remarkable and close for other sensitivity experiments of SEN-ERPs, SEN-2001 and SEN-GRASS, involving higher [PMC] decreasing in regions of DSR (average -2.9 ~ -4.5%), NCP (average -1.4% ~ -2.5%) and BTH (average -3.2% ~ -4.1%). Excluding SEN-TREE case, all the other sensitivity experiments include the forest GGW related land cover changes.

Figure 9 presents the vertical profiles of daily [PMC] and [PMC] change along the cross section (see dark solid line in Fig. 1) from 2–5 March. On 2 March, the dust plumes began to move up to atmosphere within 4 km height (Fig. 9a), and there was little [PMC] change both in DSR and NCP (Fig. 9b). On 3 March, the dust plumes were strengthened and moved to upper atmosphere about 4 km height in DSR (Fig. 9c). Some [PMC] change occurred in DSR, but not in NCP. On 4 March, the dust plumes were further strengthened and moved up to the extreme height in DSR. The [PMC] was 15–60 µg m-3 in most upper atmosphere of 4–6 km. There were strong northwest winds. Due to the strong northwest

prevailing winds, the dust plumes started to transport from upwind DSR to downwind NCP. Some dust plumes fall down to NCP, causing the [PMC] increase in NCP (see Fig. 9e and Fig. 5c). Simultaneously, obvious [PMC] change occurred in upper atmosphere of DSR. As the dust plumes fall down in NCP, some [PMC] change occurred in upper atmosphere of NCP (Fig. 9f). On 5 March, due to the strong northwest prevailing winds in the previous day, the dust plumes were blown to the upper atmosphere of NCP, with [PMC] of 15–60 μg m-3 in 4–6 km height. Meanwhile, many dust plumes dropped down to lower atmosphere of NCP, resulting in remarkable [PMC] increase in NCP, with the [PMC] rise to 100–250 μg m-3 (see Fig.9g and Fig. 5d). As the dust plumes transported to upper atmosphere of NCP, there was little [PMC] change occurred in upper atmosphere of DSR, but with remarkable [PMC] change in upper atmosphere of NCP (Fig. 9h).

There are several important issues shown in the results, and should be addressed. (1) There are heavy dust plumes during the episode, and the daily [PMC] can reach a high level in DSR and NCP. (2) The dust plumes move up to upper atmosphere and transport from upwind DSR to downwind NCP with northwest to southeast direction. (3) The vertical distribution of [PMC] decreasing accompany with the dust plumes transport.

The vertical investigations show that the dust plumes move up and evolve in upper atmosphere. However, the forest GGW will impact dust load only if dust plumes evolve in the boundary layer, because the forest acts as a barrier for dry deposition and not for emission. It is worth noting that the increased forest cover is not related to a decrease of bare surface (see Fig. 2a and Fig. S5), which means that it was not emitting dust initially. This is why the forest GGW is non-significant in dust concentration control, causing little the [PMC] decreasing (Tab. 1).

We can also found obvious hot spot of [PMC] decreasing at the crosspoints between cross section and grass GGW (see X point in Fig. 1b). As the dust plumes move up, transport and fall down, [PMC] decreasing also occurred in downwind upper atmosphere and surface (see red circle in Fig. 9b, 9d, 9f, 9h).

We find that the ERPs decrease dust erosion, concentrating on the grass GGW (Fig. S6). It is worth noting that the grass GGW is established in the edge of the dust source regions (Fig. S2c), decreasing the bare surface (Fig. 2a) with dust emitting potential, being beneficial to control dust erosion. During the episode, the total PMC emission reduction is 5.9 Gg in the research domain, illustrating that the grass GGW resultant dust erosion control is the dominant reason for [PMC] decreasing."

**In Line 516,** we added the summary of Sect. **3.4 Reasons for dust decreasing**: "4. During the episode, dust plumes move up and transport to upper atmosphere of NCP. The forest GGW is non-significant in dust concentration control, because it is benefit for dry deposition and not for emission. The grass GGW is beneficial in controlling dust erosion, being the dominant reason for [PMC] decreasing in NCP."
* * *
**52. Line 425: "dust pollutions" => "dusty episodes" Line 425: Awkward sentence: "The air pollution is severe . . . to the severe air pollutions"**

**In Line 520,** we revised the text: "dusty episodes"

**In Line 520,** we revised the text: "The air pollution is severe in eastern China, especially in NCP, and the dusty episodes have important contributions."
* * *
**53. Line 426: "ERPs help reduce some air pollutions". This is misleading. There is a clear difference between air pollution, which refers to aerosol particles produced by oxidation of anthropogenic precursors, and mineral dust particles mostly produced by wind erosion. Previous modeling study by Chin et al. (2014) shows a sharp decrease of pollution from 1990 to 2010 but an overall increase from 1980, while dust is staying pretty much constant over East Asia. You should check the entire manuscript for similar misleading definition of pollution.**

In order to make the difference, we change the dust pollution to dust concentrations or dust plumes in entire text. We added descriptions of sensitivity

experiments (see ***Response-26***). The difference between REF case and SEN-ERPs case denotes the different of model results with or without GGWs (grass GGW and forest GGW), excluding the impacts of near-surface wind speed, the surface wetness and other factors. Compared to the sensitivity experiments, the REF case is the base case with grass GGW and forest GGW, revealing lower [PMC] during the dust episode. So the "ERPs help to reduce some air concentrations" here only denotes the influence of ERPs on dust plumes in NCP during the dust storm episode, revealing no inconsistency with the study of Chin et al., 2014.

To express clearly, we revised the text added the description about the study of Chin et al., 2014.

**In Line 512,** we description of Chin et al., 2014 the text: "It worth noting that the "ERPs help to reduce some dust concentrations" here only denotes the states with or without GGWs, revealing no inconsistency with the study of Chin et al., 2014, who have found dust is staying pretty much constant over East Asia."

**In Line 5120,** we revised the text resultant dust emission decreasing: "This study shows that ERPs induced remarkable vegetation increase, especially the grass GGW, being beneficial in controlling dust erosion, reducing dust concentrations in NCP, especially in springtime."

**In Line 522,** we revised the text: "The air pollution is severe in eastern China, especially in NCP, and the dusty episodes have important contributions, illustrating the considerable beneficial of ERPs to air pollution control in China."
* * *
**54. Lines 430-432: Awkward sentence. Please reformulate and do not use the word "sketchy" to characterize your work. I don't think that Atm. Chem. Phys. will publish "sketchy" work.**

**In Line 524,** we removed "sketchy" and revised the text: "It should be reiterated that, considering the limitation of case study, the main focus of this study do not intent to give a general conclusion, but rather to provide some insights of the effect of ERPs on the land cover change and resultant decreasing of dust

concentration over downwind areas, where heavy haze often occurred due to anthropogenic air pollutants."
* * *
**55. Line 655: "barrens"=>"bare soil" or "bare surface" and change in the Figure**

We replaced "barren" by "bare soil" for related text and figures.
* * *
**56. Figure 3: What is the X-axis: day, month, and year? What are the units for the Y- axis? Replace "The model performance statistics o NMB and IOA are also shown" by providing the full name of NMB and IOA (I even cannot find it in the text!).**

In Figure 3, X-axis is day on March 2016 in Beijing Time. The unit of Y-axis is $\mu g\ m^{-3}$.

We revised the X-axis in Figure 3.

We revised the figure caption.
* * *
**57. Figure 4 caption: What is the period covered by "episode average"? Figure 4 b is not a "correlation analysis". Define circles and lines.**

The period coverage by "episode-average" is from 2 to 8 March 2016.

We revised Figure 4 and related caption.

In Line 328, we revised the related text: "The episode-averaged model results were compared with the measurements in Fig. 4."
* * *
**58. Figure 5 caption: "The correlation indices (R) between measurements and simulations are also presented"=> "The correlation coefficient (r) of the linear regression between simulated and observed surface concentration is indicated in red."**

We revised the caption of Figure 5.
* * *
**59. Figure 6 caption: Need major improvement, as it is impossible to know what is shown on this Figure from the caption**

To describe more clearly, we dived Figure 6 into two Figures. One is for spatial variations (Fig.6) and the other is for temporal variations (Fig. 7). The descriptions of experiments are detailed in *Response-46*.
* * *
**60. Figure 7: Right panels are not defined properly as I have no clue what they are. The entire caption needs improvement for clarity.**

We revised the caption of Figure 7 (**Figure 8** in revised version).
* * *
**In the revised text, we added references:**

Chin, M., Ginoux, P., Kinne, S., Torres, O., Holben, B. N., Duncan, B. N., Martin, R. V., Logan, J. A., Higurashi, A., and Nakajima, T.: Tropospheric Aerosol Optical Thickness from the GOCART Model and Comparisons with Satellite and Sun Photometer Measurements, J. Atmos. Sci., 59, 461--483, 2002.

Chin, M., Diehl, T., Tan, Q., Prospero, J. M., Kahn, R. A., Remer, L. A., Yu, H., Sayer, A. M., Bian, H., and Geogdzhayev, I. V.: Multi-decadal aerosol variations from 1980 to 2009: a perspective from observations and a global model, Atmos. Chem. Phys., 14, 3657-3690, 2014.

Chou, M. D., and Suarez, M. J.: A solar radiation parameterization for atmospheric studies, Nasa Tech.

Rep., NASA/TM-1999-104606, , 15, 38 pp., 1999.

Chou, M. D., Suarez, M. J., Liang, X. Z., Yan, M. H., and Cote, C.: A Thermal Infrared Radiation Parameterization for Atmospheric Studies, NASA/TM-2001-104606, 19, 55, pp., 2001.

Fang, J., Chen, A., Peng, C., Zhao, S., and Ci, L.: Changes in forest biomass carbon storage in China between 1949 and 1998, Science, 292, 2320, 2001.

Ginoux, P., Joseph, P. M., Thomas, G. E., Hsu, C. N., and Zhao, M.: Global scale attribution of anthropogenic and natural dust sources and their emission rates based on MODIS Deep Blue aerosol products, Rev. Geophy., 50, 3005, 2012.

Grousset, F. E., Ginoux, P., Bory, A., and Biscaye, P. E.: Case study of a Chinese dust plume reaching the French Alps, Geophys. Res. Lett., 30, 10-11, 2003.

Kalnay, E., Kanamitsu, M., Kistler, R., Collins, W., Deaven, D., Gandin, L., Iredell, M., Saha, S., White, G., and Woollen, J.: The NCEP/NCAR 40-Year Reanalysis Project, B. Am. Meteorol Soc., 77, 437-472, 1996.

Miller, R. L., and Tegen, I.: Climate Response to Soil Dust Aerosols, J. Climate, 11, 3247-3267, 1998.

Silva, L. C. R., and Anand, M.: Probing for the influence of atmospheric $CO_2$ and climate change on forest ecosystems across biomes, Global Ecol. Bioge., 22, 83-92, 2013.

Su, X. L., Wang, Q., Li, Z. Q., Calvello, M., Esposito, F., Pavese, G., Lin, M. J., Cao, J. J., Zhou, C. Y., Li, D. H., and Xu, H.: Regional transport of anthropogenic pollution and dust aerosols in spring to Tianjin - A coastal megacity in China, Sci. Total Environ., 584, 381-392, 10.1016/j.scitotenv.2017.01.016, 2017.

Tegen, I., Werner, M., Harrison, S. P., and Kohfeld, K. E.: Relative importance of climate and land use in determining present and future global soil dust emission, Geophys, Res, Lett,, 31, 325-341, 2004.

Wu, W., Shibasaki, R., Ongaro, L., Ongaro, L., Zhou, Q., and Tang, H.: Validation and comparison of 1 km global land cover products in China, International Journal of Remote Sensing, 29, 3769-3785, 2008.

---

## Author Comment (AC5) · 9 Feb 2018

**Response to Referee #1**

We thank the reviewers for the careful reading of the manuscript and helpful comments. According to the suggestions of the reviewer, the reviewers' comments have been carefully addressed, and the paper is carefully revised. We believe that the revised paper has been significantly improved after addressing the comments of the reviewers.

\*\*\*\*\*\*\*\*\*\*\*\*\*\*\*\*\*\*\*\*\*\*\*\*\*\*\*\*\*\*\*\*\*\*\*\*\*\*\*\*\*\*\*\*\*\*\*\*\*\*\*\*\*\*\*\*\*\*\*\*\*\*\*\*\*\*\*\*\*\*\*\*\*\*

**This manuscript provides a case study of changed landuse fraction on the dust storm over Northern China. Its method is straightforward and easy to understand. My main concern is whether the single case study of 5 days is sufficient for the climatological pattern shift of dust storm as the paper title states. The authors may consider study more cases for more years. For instance, this single case shows that the dust storms strength became weaker after changing its landuse. How about the frequency of the dust storm occurrence for one or more year? It would be better to add more convincing cases.**

We agreed the reviewer that the single case cannot provide a general conclusion, but it provides some important insights of ERPs' effects.

1) We described the limitation of previous studies, which could implicated the insights and importance of our work **in Line 108**: "The previous studies didn't quantify the roles of ERPs on dust concentrations, such as the detailed land cover change induced by ERPs, the effect of regional dust transport to downwind regions, especially in the NCP region, etc."

2) We emphasized the theme of our work with "a case study" **in Title**: "**Effect of ecological restoration programs on dust concentrations in North China Plain: a case study**".

3) We also reiterated the limitation of our work in the summary and conclusions **in Line 524**: "It should be reiterated that, considering the limitation of case

study, the main focus of this study do not intent to give a general conclusion, but rather to provide some insights of the effect of ERPs on the land cover change and resultant decreasing of dust concentration over downwind areas, where heavy haze often occurred due to anthropogenic air pollutants."

4) In order to address the reviewer's concern, we added a new case simulation in different year from 22 to 26 May 2014. We have conducted another simulation from 22 to 26 May 2014 to investigate the influence of ERPs to the dust concentrations in NCP. This simulation shows that the EPRs help to reduce the dust particle concentrations from -5% to -15% in BTH, NCP, and DSR, respectively (see Fig. S7). This result is similar to the case in 2016. Because the frequency of the dust storm occurrences is different in different years, this new simulation shows some evidences that the ecological restoration programs in China plays important roles to reduce the dust concentrations in eastern China. More detailed discussion can be seen in the Supplementary **Section SI-1**: **Effect of ERPs on dust concentrations in NCP during another dust events from 22 to 26 May 2014.**

"The model simulations from 2 to 7 March 2016 show that the EPRs help to reduce the dust concentrations in NCP, especially in BTH, involving [PMC] reduction ranges from -5% to -15%. In order to further confirm the important role of ERPs transport, another dust events from 22 to 26 May 2014 in NCP is simulated using the WRF-DUST model.

Figure S6 shows the daily average calculated and measured [PMC] distributions. On 22 to 23 May 2014, the dust storm was started and strengthened in DSR region, both the observed and simulated [PMC] reached as high level in the upwind DSR region, while with low value (lower than 40 µg m$^{-3}$) in the downwind NCP region (Fig. S6a, S6b). On 24 May, the dust storm started to be transported from upwind DSR to downwind NCP with northwest to southeast direction due to the strong northwest prevailing winds (Fig S6c). On 25 May, the dust storm reached to the NCP region, and caused a remarkable [PMC] increase, rising to 100–250 µg m$^{-3}$. On 26 May, the dust storm passed through and the wind speed slowed down, the [PMC]

significantly decreased in NCP region (Fig. S6e). The correlation coefficients between measured and simulated [PMC] are 0.66–0.87 during the episode (Fig. S6). Despite some model biases, the WRF-DUST model well captures the evolutions of dust storm during 22 to 26 May 2014.

Figure S7 presents the hourly near-surface [PMC] change during the dust events from 22 to 26 May 2014, including the temporal variations in concentrations and percentage averaged at monitoring sites in the regions of DSR, NCP and BTH. During the episode when the dust storm was transported from DSR to NCP, the [PMC] reduction induced by ERPs performs with the maximum reduction of [PMC] ranging -5% to -15% in NCP. The results suggest that ERPs decrease the dust concentrations in NCP, which is consistent with the previous dust events during 2 to 7 March 2016 (Tab. S2)."